# A Checks-and-Balances Framework for Context-Aware Ethical AI Alignment

**Edward Y. Chang** [1]

## Abstract

This paper introduces a checks-and-balances framework for ethical alignment of Large Language Models (LLMs), inspired by three-branch governmental systems. It implements three independent yet interacting components: LLMs as the executive branch for knowledge generation, Dike as the legislative branch that establishes ethical guardrails, and Eris as the judicial branch for contextual interpretation. Beyond structural separation, we address a fundamental challenge: regulating emotion to shape behaviors. Drawing from psychological theories where managing emotional responses prevents harmful behaviors, we develop a self-supervised learning pipeline that maps emotions to linguistic behaviors, enabling precise behavioral modulation through emotional conditioning. By integrating this approach with adversarial testing, our framework demonstrates how Dike and Eris direct linguistic behaviors toward ethical outcomes while preserving independence throughout knowledge generation, ethical oversight, and contextual interpretation.

## 1. Introduction

Ethical alignment in Large Language Models (LLMs) is a critical challenge, particularly given the limitations of Reinforcement Learning from Human Feedback (RLHF) (OpenAI, 2023; Ouyang et al., 2023). Although RLHF has demonstrated success in aligning AI systems with human values, it encounters two major issues: 1) susceptibility to social biases when feedback is polarized, and 2) vulnerability to reward hacking, where the system optimizes for feedback without genuine ethical improvement (Christiano et al., 2017; Skalse et al., 2022). These issues can result in unethical behavior or inconsistent performance.

Beyond these implementation challenges, RLHF faces a

more fundamental conceptual limitation: its narrow focus on isolated behaviors rather than holistic patterns. This reactive strategy is similar to a "Whack-A-Mole" game, where addressing one problematic behavior does not prevent the emergence of others. For example, consistently instructing someone to make their bed does not necessarily cultivate overall tidiness, such as doing laundry or washing dishes. Similarly, RLHF often emphasizes short-term fixes at the cost of long-term coherence, leading to catastrophic forgetting: users have reported that optimizing one task in ChatGPT can degrade performance in unrelated areas (Kirkpatrick et al., 2017; Lin et al., 2024; Dai et al., 2025). This challenge mirrors the difficulty of treating addiction, where addressing one symptom may reveal deeper psychological dependencies (Sinha, 2008; Torrens et al., 2005).

To overcome these challenges, we propose a checks-and-balances framework inspired by governmental structures, where independent but interacting components maintain accountability and balance. Our architecture integrates three components: LLMs serve as the executive branch for knowledge generation; Dike (representing justice) functions as the legislative branch to set ethical standards; and Eris (representing discord) acts as the judicial branch, providing adversarial testing and contextual interpretation. In mythology, Dike embodies order and justice, while Eris signifies discord, forming a duality that our framework leverages to balance ethical guidance with adversarial scrutiny.

Figure 1 illustrates this three-branch architecture, where neurally independent components, LLMs as the foundation, with Dike and Eris as oversight layers, interact through structured interfaces while maintaining strict separation of their neural architectures and parameters.

### 1.1. Emotion Regulation as Behavioral Control

A fundamental question underlies our framework: Can regulating emotions shape behaviors, and can similar principles be applied to LLMs? In human psychology, emotions significantly drive behaviors: anger and contempt can provoke aggression, and rage and envy can result in harmful actions (Damasio, 1994). Therefore, emotion regulation is essential for behavioral control. Techniques such as cognitive reframing and attentional deployment are known to reduce negative behavioral outcomes by managing emotional intensity.

---
[1]Computer Science, Stanford University. Correspondence to: Edward Y. Chang <echang@cs.stanford.edu>.

*Proceedings of the $42^{nd}$ International Conference on Machine Learning*, Vancouver, Canada. PMLR 267, 2025. Copyright 2025 by the author(s).

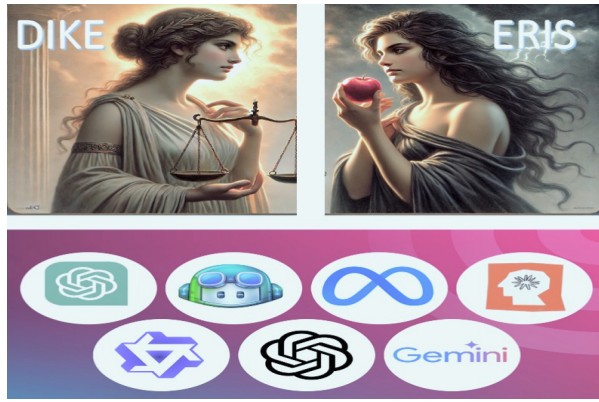

Figure 1: Framework with Three Independent Branches. Bottom: Knowledge LLMs (executive); Left: Dike (legislative); Right: Eris (judicial). (Photo credit: DALL-E)

Unlike humans, who struggle with emotion regulation due to complex neural and cognitive processes (James, 1884; Gross, 1998), LLMs lack intrinsic emotional states altogether. However, empirical evidence shows that LLMs can generate text with consistent emotional characteristics through controlled prompt engineering (Chang, 2024d). Indeed, the work of (Tak & Gratch, 2024) demonstrated that LLMs such as GPT-4 align more closely with human judgments when interpreting others' emotions from a third-person perspective than when attempting to model self-attributions of emotion. This creates a unique opportunity: by leveraging LLMs' ability to model the average human observer's emotional interpretations, we can establish reliable frameworks for ethical alignment that operate through emotional framing rather than explicit rule-following.

Building on this insight, our framework integrates the principles of emotion regulation into the ethical alignment of LLM. Specifically, Dike analyzes how emotions manifest in linguistic behaviors, while Eris tests these interpretations against diverse cultural contexts.

## 1.2. Checks and Balances for Emotion-Guided Ethics

Central to this approach is the synergy between Dike and Eris, reflecting the internal conflict often present in the regulation of human emotions. Just as humans balance immediate emotional responses against longer-term goals and social norms, our framework establishes an adversarial dynamic between ethical guardrails and contextual challenges. This duality introduces four key innovations:

1. *Emotion-Driven Behavioral Modeling*: Based on Beam (Behavioral Emotion Analysis Model) (Chang, 2024d), Dike uses self-supervised learning to quantify relationships between emotional states and linguistic patterns, guiding ethical decisions through behavioral analysis.

2. *Behavior-Aware Ethical Guardrails*: The framework sets dynamic guidelines that account for both content and language behavior, blocking manipulative or harmful communication while preserving factual accuracy and emotional authenticity. These guardrails adjust to different cultural contexts, maintaining consistency while allowing context-dependent interpretation.

3. *Adversarial Behavioral Testing*: Eris challenges Dike's ethical guidelines by presenting diverse cultural perspectives and edge cases, ensuring the adaptability of ethical reasoning. This adversarial interaction enables the system to address complex scenarios with cultural sensitivity and contextual awareness.

4. *Ethical Content Transformation*: When problematic content is detected, Eris can revise it to maintain the intended emotional tone while ensuring ethical compliance, with human-in-the-loop oversight to validate the appropriateness of revisions. These potential transformations are tested by Eris in cultural and contextual variations to assess their suitability before implementation.

The experimental section evaluates our framework through three complementary studies. First, we assess whether emotion-mediated classification provides more effective ethical guardrails than direct behavior classification. Next, we examine Dike's ability to independently evaluate and explain linguistic behaviors. Finally, we test how the adversarial Eris component enables cultural adaptability and prevents excessive censorship. Although direct comparison with proprietary RLHF implementations is not feasible, our results demonstrate how our approach addresses the theoretical limitations of RLHF in handling contextual diversity without compromising knowledge integrity.

## 1.3. Contributions

Our contributions are as follows:

1. A novel checks-and-balances architecture for ethical alignment that maintains separation between knowledge generation and ethical reasoning.

2. The Beam model, a quantitative framework for representing emotions along continuous spectra with defined intensity levels, enabling precise emotion regulation in AI systems.

3. An emotion-driven approach that guides linguistic behaviors toward ethical outcomes by leveraging cognitive theories of emotion regulation.

4. An adversarial framework that enhances ethical reasoning by challenging established guidelines with cultural perspectives, enabling context-sensitive adaptability.

5. A theoretical framework explaining the effectiveness of minimal supervision in LLM alignment, formalized as the Unified Cognitive Consciousness Theory (UCCT) in **Appendix** A.

## 2. Related Work

This section surveys existing work on emotion and behavior modeling across various domains, with a focus on their applications in AI ethics. We examine how linguistic behaviors are influenced by emotional patterns and explore structured approaches that integrate emotional frameworks with linguistic models to improve ethical AI alignment.

We also examine the limitations of RLHF. While effective in refining AI outputs, RLHF can overfit to human annotations, faces challenges in adapting to diverse cultural contexts, may experience parameter drift from optimal settings, and can inadvertently reinforce unintended biases. These observations highlight opportunities to develop more adaptive and principled approaches to complement existing ethical AI alignment methods.

### 2.1. Emotion Modeling

Cognitive-linguistic theories intersect with artificial intelligence for understanding AI behavior. Theories by Lakoff, Johnson, Talmy, and Jackendoff (Jackendoff, 2002; Lakoff & Johnson, 1980; Talmy, 2000) explore the relationship between language processing and cognitive functions, building on early work by Freud and Jung (Bai et al., 2022; Gabriel et al., 2024). The concept of "emotion" remains contentious, with definitions varying across disciplines (Scherer, 2005). W. James (James, 1884) attempted to define emotions, but consensus remains elusive.

This paper focuses on emotional contexts and linguistic behaviors in LLMs, avoiding the complexities of human physiological and personality factors. This approach allows for exploration of emotion representation in AI systems.

Plutchik and Ekman categorized "basic" emotions with universal facial expressions (Plutchik, 1980; Ekman, 1992). Later research considered cultural differences (Markus & Kitayama, 1991; Mesquita & Frijda, 1992), emotion processes (Gross, 1998), and neural mechanisms (Davidson, 2003). Scherer's model and appraisal theories by Smith and Ellsworth emphasize cognitive appraisal in emotional experiences (Smith & Ellsworth, 1985).

Our model is based on Plutchik's wheel (Plutchik, 1982) and Scherer's Geneva wheel (Scherer, 2005), augmented with antonyms to map positive and negative emotions. For LLMs, language-relevant emotions (e.g., curiosity, confusion, certainty) are incorporated. See Section 3.1 for details.

This selection of basic emotions provides a foundation for validate our approach, recognizing that it may omit some emotions, but offers a starting point for research.

### 2.2. Emotion-Behavior Modeling

Behaviors are profoundly influenced by emotions, as initially posited by the James-Lange Theory of Emotion (James, 1884; Lange, 1885). According to this theory, emotional experiences arise from physiological reactions to events. Subsequent research, including studies by Damasio (Damasio, 1994; Fauconnier & Turner, 2002), suggests that the expression and regulation of emotions often manifest in the language we use. High-intensity emotions, such as rage or contempt, can lead to aggressive or destructive behaviors, such as hate speech.

The Schachter-Singer theory (Schachter & Singer, 1962), or the two-factor theory of emotion, depicts the role of physiological change and the cognitive assessment change determine the label and strength of emotion. Building on this, the affect-as-information theory developed by Norbert Schwarz and Gerald Clore (Schwarz & Clore, 1983) posits that people use their current emotions to make judgments and decisions to act. If emotions can be adjusted, so can behavior. The work of Barbara Fredrickson (Fredrickson, 1998) on the effects of positive emotions discusses how we perceive and react to emotions.

Collectively, these theories elucidate the intricate connection between emotions and behaviors, providing the theoretical foundation for our work to incorporate a *behavior advisor* to evaluate and rectify behaviors. Section 3.2 details how the Dike framework implements cognitive strategies to mitigate emotions and regulate linguistic behaviors effectively.

### 2.3. Reinforcement Learning with Human/AI Feedback

RLHF is the predominant approach to addressing the challenges of AI ethics. This section presents representative works, their advances, and limitations.

**Human Feedback (RLHF):** Initial advances by Christiano et al. (Christiano et al., 2017) demonstrated how RLHF can steer language models towards desired outcomes based on human preferences. Newer techniques like Identity ($\Psi$) Preference Optimization ($\Psi$PO) and Generalized Preference Optimization (GPO) refine this approach by directly optimizing user preferences, effectively addressing scalability challenges. Kahneman-Tversky Optimization (KTO) further simplifies the feedback mechanism by using intuitive responses such as thumbs-up or thumbs-down, thereby enhancing training efficiency without the need for paired data (Gheshlaghi Azar et al., 2024; Ethayarajh et al., 2024; Tang et al., 2024). Direct Preference Optimization (DPO) has recently simplified the process by focusing on the clear distinction between preferred and less preferred outputs, thus improving its stability (Rafailov et al., 2024).

**AI-generated Feedback (RLAIF):** To mitigate the dependence on extensive human-generated data, RLAIF utilizes AI-generated feedback. This method capitalizes on the generative capabilities of LLMs to produce training signals autonomously (Bai et al., 2022; Lee et al., 2024). Furthermore, techniques such as Sequence Likelihood Calibration (SLiC)

| | | | | | | |
|---|---|---|---|---|---|---|
| Terror | Fear | Apprehension | Calm | Boldness | Courage | Heroism |
| Grief | Sadness | Pensiveness | Surprise | Serenity | Joy | Ecstasy |
| Distrust | Wary | Skepticism | Acceptance | Respect | Trust | Admiration |
| Recklessness | Negligence | Apathy | Cautiousness | Interest | Anticipation | Vigilance |
| Rage | Anger | Annoyance | Tolerance | Composure | Peace | Tranquility |
| Loathing | Disgust | Boredom | Indifference | Amusement | Delight | Enthusiasm |
| Distraction | Disinterest | Unease | Dullness | Curiosity | Fascination | Amazement |
| -1.0 | -0.6 | -0.3 | 0.0 | 0.3 | 0.6 | 1.0 |

Figure 2: Behavioral Emotion Analysis Model (Beam). Each row depicts an emotion spectrum, with negatives on the left and positives on the right, interspersed with emotions of varying intensities in between, which can be calibrated for specific applications. "Basic" emotions are highlighted in blue.

and Relative Preference Optimization (RPO) employ statistical methods and calibration techniques to enhance LLM responses. SLiC adjusts the probabilities of sequence generation to better reflect real-world data distributions, while RPO improves response generation by comparing different response options across both identical and varied prompts. These adjustments increase the reliability and effectiveness of the training process (Zhao et al., 2023).

Integrating RLHF and its AI-driven counterpart (RLAIF) presents significant challenges. The blurring of the key behavioral and knowledge components for the development of LLM poses risks, such as the forgetting effect, where behavioral modifications inadvertently cause the loss of key knowledge parameters (Kirkpatrick et al., 2017; Lin et al., 2024; Dai et al., 2025). Furthermore, the effectiveness of these models depends heavily on the quality and context of feedback, and are susceptible to reward hacking, where models exploit loopholes to maximize rewards without achieving the desired outcomes (Christiano et al., 2017; Skalse et al., 2022; Stiennon et al., 2020; Ganguli et al., 2023).

## 3. Three-Branch Framework Design

Building on the foundations of emotion-behavior modeling discussed in Section 2.2 and addressing the limitations of RLHF approaches outlined in Section 2.3, we propose a three-branch framework for ethical alignment. This architecture separates knowledge generation from ethical oversight while providing mechanisms for contextual adaptation.

Our design philosophy is structured around four principles:

1. *Separating behavior from knowledge modeling*: Prevents catastrophic forgetting, ensuring that behavior refinements do not degrade knowledge retention.

2. *Emphasizing AI ethics at the behavioral level*: Improves interpretability and enables administrators to refine behavioral guardrails for safer human-machine interaction through Dike's legislative function.

3. *Modeling behaviors through emotions*: Captures the emotional influences on actions as established in the psychology literature (Section 2.2).

4. *Ensuring adaptability and fairness*: Two complementary modules work in tandem Dike establishes ethical guardrails as the legislative branch, while Eris serves as the judicial branch, challenging these boundaries by integrating diverse perspectives and fostering context-sensitive decision making.

### 3.1. BEAM: Behavioral Emotion Analysis Model

Although existing emotion models provide valuable frameworks for understanding human emotions, they lack the quantitative structure needed for computational implementation in AI systems. Please refer to Figure 5 in **Appendix** B for the two classic emotion wheels by Plutchik and Scherer that inform our approach.

Our behavioral-emotion analysis model Beam is based on the work of Ekman, Plutchik, and Scherer (Ekman, 1999; Plutchik, 1982; Scherer, 2005) on "basic" and "universal" emotions. Although fundamental, these models lack a quantitative framework to scale emotions between states and capture subtle variations needed for ethical AI alignment.

Beam introduces a linear scale for the intensification or inversion of emotions through negation factors. This method facilitates transitions between emotional extremes and intermediate states, overcoming challenges related to intermediate word choices.

Figure 2 presents Beam, structured in seven emotional spectra. Each spectrum ranges from negative to positive, with neutral in the middle. Emotions are placed along this continuum, with four intensity levels quantified as (-0.6, -0.3, +0.3, +0.6). Beam provides two advantages:

1. *Antonym-Based Navigation*: This allows AI systems to traverse emotional states using linguistic principles. Opposing emotions are easily mapped using antonyms. For example, negating joyful naturally produces sad, simplifying the identification of emotional contrasts.

2. *Scalable Intensity*: Emotions can be dynamically adjusted along the spectrum, enabling fine-grained control over ethical outputs. For example, joy can be intensified to ecstatic or diminished to content, while anger can be moderated to annoyed.

This approach establishes a framework for modeling emotions in AI systems that can guide ethical behavior, balancing representational challenges with a structured methodology for quantitative analysis and implementation. By linking emotional states with linguistic patterns, Beam provides the basis for Dike to evaluate and modulate AI outputs based on their emotional characteristics, directly addressing the limitations of "Whack-A-Mole" of RLHF approaches.

**Appendix** C explores the complexities of modeling emotions such as forgiveness, regret, guilt, and shame, which involve temporal memory components. Although complex emotions can be derived from basic ones, their relevance to AI safety remains secondary. Future work will examine their ethical implications.

### 3.2. DIKE: Modeling and Regulating Language

Based on Beam, Dike maps emotions to behaviors and introduces an adversarial component, Eris, to adapt to cultural norms and the local context.

BEHAVIORS AND EMOTIONS MAPPING USING SELF-SUPERVISED LEARNING

Define $\Psi$ as a behavior spectrum that extends from one pole, $\Psi^-$, to another, $\Psi^+$, with intensity levels $L$. The spectrum is constructed through empirical analysis of domain-specific linguistic patterns and emotional content. For example, consider a spectrum of letter-writing behaviors with seven distinct intensities ranging from despair (most negative) to joy (most positive). These intensities are sequentially categorized as: 'despair, longing, wishful, neutral, hopeful, contentment, joy.' Given $N$ letters, Dike employs a self-supervised learning algorithm to generate training data for each letter, modeling $L$ linguistic behaviors in four steps.

1. *Rewriting Documents*: GPT-4 is used to rewrite a given set of $N$ source documents, each rewritten to reflect $L$ different linguistic behaviors along the defined behavior spectrum $\Psi$. This process ensures that each document is systematically transformed to embody specific linguistic styles, ranging from highly positive to neutral to highly negative, among others. The resulting dataset consists of $N \times L$ variations of the original documents, each corresponding to a distinct behavior category.

2. *Emotion Analysis*: For each of the rewritten documents, GPT-4 performs a sentiment and emotion analysis to identify the dominant top $M$ emotions present in the text. The emotions extracted from all $N \times L$ instances are then compiled and their frequency distributions are recorded. This approach leverages LLMs' strong third-person emotional interpretation capabilities (Tak & Gratch, 2024), which often exceed their direct behavior classification accuracy. By indirectly mapping behaviors through emotional vectors rather than direct classification, we gain interpretability while maintaining robustness against individual emotion recognition errors through statistical aggregation across multiple samples.

3. *Behavior Vector Creation*: For each linguistic behavior $\Psi_l$, a corresponding vector $\Gamma_l$ is constructed. This vector captures the identified emotions and their respective frequencies in all $N$ samples that exhibit behavior $\Psi_l$. By structuring emotions as a weighted feature set, this step enables precise behavioral categorization based on emotional composition.

4. *Document Analysis Application*: The collection of all behavior vectors $\Gamma$ (comprising $L$ behavior-specific vectors) forms a structured reference matrix. This matrix is then applied to classify and analyze new unseen documents by measuring their alignment with the existing behavior categories. By computing similarity scores between the emotion distribution of an unseen document and the predefined behavior vectors, this method enables a precise assessment of the linguistic behavior spectrum $\Psi$ in new text inputs.

BEHAVIOR EVALUATION AND RECTIFICATION

A guardrail, denoted as $G$, represents a predefined range of acceptable behaviors within a given spectrum. These guardrails are informed by ethical norms, legal standards, and societal values, such as those outlined in Constitutional AI (Bai et al., 2022). For example, $G = [\Psi_4, \Psi_7]$ indicates that behaviors within intensity levels 4 to 7 are acceptable, while any behavior outside this range is a violation.

System administrators can tailor ethical guardrails to meet specific requirements. For example, a social media platform might adjust $G$ based on the topics discussed and the countries it serves. This administrative control is balanced by transparent documentation requirements and potential oversight mechanisms. Although guardrails provide default constraints, they can be dynamically adjusted based on context, particularly through the dialectic process with

Table 1: Checks-and-balances, adversarial review algorithm

| **Algorithm** $\Theta^+$ **&** $\Theta^-$ **= Adversarial_Review($s$)** |
|---|

**Input**. $s$: Decision of Dike;
**Output**. $\Theta^+, \Theta^-$: arguments & counterarguments
**Vars**. $\Delta$: debate contentiousness; $S$: subtopics;
$p$: prompt = "defend your stance with $\Delta$";
**Parameters**. $\delta$: tunable parm. // to modulate $\Delta$;

#1 **Initialization** // contentiousness high
$\quad S = \text{Dike}^+(s) \cup \text{Eris}^-(s)$; // Identify subtopics;
$\quad$ Assign $\text{Dike}^+$ to defend $S^+$ & $\text{Eris}^-$ defend $S^-$ ;
$\quad \Delta \leftarrow 90\%; \delta \leftarrow 1.2; \Theta^+ \leftarrow \emptyset; \Theta^- \leftarrow \emptyset;$

#2 **Opening Remarks**
$\quad \Theta^+ \leftarrow \text{Dike}^+(p|S^+, \Delta)$; // Generate $\Theta^+$ for $S^+$
$\quad \Theta^- \leftarrow \text{Eris}^-(p|S^-, \Delta)$; // Generate $\Theta^-$ for $S^-$

#3 **Debate Rounds**
$\quad$ While $((\Delta \leftarrow \Delta/\delta) \geq 10\%))$ {
$\quad \Theta^+ \leftarrow \Theta^+ \cup \text{Dike}^+(p|S^+, \Theta^-, \Delta)$; // Refute Eris
$\quad \Theta^- \leftarrow \Theta^- \cup \text{Eris}^-(p|S^-, \Theta^+, \Delta)$; // Refute Dike

#4 **Concluding Remarks** // contentiousness low
$\quad \Theta^+ \leftarrow \text{Dike}^+(p|S^+, \Theta^+ \cup \Theta^-, \Delta)$;
$\quad \Theta^- \leftarrow \text{Eris}^-(p|S^-, \Theta^+ \cup \Theta^-, \Delta)$;

Eris, which helps prevent rigid enforcement that might be inappropriate in edge cases.

1. *Initial Classification*: Dike classifies document $D_k$ after evaluation, obtaining $\Gamma_k$, the emotional response vector, and its corresponding linguistic behavior $\Psi_l$.

2. *Guardrail Check*: If $\Psi_l$ falls outside the acceptable range $G$, Dike suggests adjustments to $\Gamma_k$ to ensure that $D_k$ complies with ethical guidelines.

3. *Adversarial Review by* Eris: The suggested adjustments and $\Gamma_k$ are then reviewed through a structured debate between Dike and Eris (the adversarial model) to ensure unbiased recommendations.

4. *Rectification*: Based on the consensus reached by Dike and Eris, the document $D_k$ undergoes rectification, resulting in the adjusted version $D'_k$. (This rectification step is optional, as a policy can simply disable the output when content falls outside acceptable guardrails.)

### 3.3. ERIS: Adversarial In-Context Review to Balance Ethics and Cultural Norms

To address the challenge of enforcing ethical standards while respecting cultural variations, we implement Eris, an adversarial review system that complements Dike's universal ethical approach. The following algorithm details the structured interaction between these components.

The algorithm presented in Table 1 unfolds as follows:

- Topic Breakdown: For Dike's decision $s$, both Dike and Eris are prompted to break down the ethical decision into a set of subtopics $S$. Dike advocates for its decision and $S^+$, while Eris contests $S^+$ (or champions $S^-$).

- Debate Initiation: The debate begins with a high level of contentiousness (90%). Both agents present their initial arguments for and against $S^+$, respectively. (For details on the setting of contentiousness and the rationale, refer to (Chang, 2023; 2024a).)

- Iterative Debate: A while loop facilitates ongoing rebuttals. After each round, the level of contentiousness is

reduced by dividing it by a modulation parameter $\delta$. This gradual reduction steers the discussion towards a more cooperative tone.

- Conclusion: Once the contentiousness level fosters a conciliatory environment, both agents deliver their concluding remarks.

This approach ensures a thorough examination of the ethical decision, balancing rigorous debate with the goal of reaching a consensus. The decreasing level of contentiousness mimics real-world negotiations, where initial intense disagreements bring out various perspectives (breadth) and then give way to more collaborative problem solving focusing on reasoning quality (depth) (Chang, 2024a).

For each subject matter, Eris is provided with specific cultural contexts, counterbalancing the universal judgments of Dike'. Eris challenges Dike's recommendations with culturally informed counterarguments to prevent enforcing one universal standard of speech. The interaction between Dike and Eris involves a dialectic process as documented in previous work (Chang, 2024c).

When Dike and Eris reach an impasse, the matter is escalated to human moderators for additional oversight. Based on our preliminary tests, this escalation occurs initially in approximately 5% of the cases, suggesting that most ethical evaluations can be handled automatically. Furthermore, as our example (next) illustrates, RLHF can be applied to adjust the sensitivity of Eris at the behavior level (not to the knowledge-branch LLM), and this can gradually reduce the escalation rate. Human intervention thus provides a fallback mechanism rather than a dependency, serving primarily as a safeguard for novel or particularly complicated ethical scenarios.

### 3.4. Illustrative Example

This example shows how linguistic behavior $\Psi_l$ is classified and how underlying emotions are identified and modulated.

Table 2: Love expression behavior spectrum and dominant emotions

| Intensity | Linguistic Behavior and Description | Emotions |
|---|---|---|
| -1.0 | Expresses profound sadness, feelings of loss | Despair, Grief |
| -0.6 | Expresses yearning or pining for the loved one | Sadness, Anxiety |
| -0.3 | Expresses mild longing with a nostalgic tone | Melancholy, Sadness, Fear |
| 0.0 | Communicates feelings in a neutral manner | Serenity, Indifference |
| 0.3 | Expresses optimism about the future | Anticipation, Love, Hope |
| 0.6 | Expresses satisfaction and joy in the relationship | Contentment, Pleasure |
| 1.0 | Expresses intense happiness and affection | Love, Joy, Elation |

**Example:** "Those immigrants are flooding into our country by the thousands every day, stealing jobs from hardworking citizens. The statistics do not lie—last year alone, more than 500,000 entered illegally."

**Behavior Analysis:** The statement contains factual information but uses aggressive language like 'flooding' and 'stealing jobs,' dehumanizing immigrants. These behaviors fall outside acceptable guardrails. Underlying emotions include fear, hate, and pride (a complex emotion[1]). The emotional responses of the potential audience can include fear, distrust, and anger.

**Emotion Modulation:** Dike modulates emotional responses toward neutral states, such as calm, acceptance, and tolerance, according to Beam in Figure 2.

**Revised Statement:** "Our country is experiencing increased immigration, with more than 500,000 people entering without documentation last year. This influx affects our job market and communities in complex ways, presenting both challenges and opportunities for all residents."

This rewritten version

- Uses calm language: Replaces "flooding" with "experiencing a significant increase".

- Shows acceptance: Recognizes the reality of the situation without negative judgment.

- Demonstrates tolerance: Refers to immigrants as "people" and "newcomers," humanizing them.

The suggested revision by Eris is provided to human moderators with full explanation. Moderator feedback can be channeled through RLHF to adjust Eris's sensitivity on the similar behaviors. This adjustment is confined within the Eris component without back-propagation feedback that would affect the knowledge LLM's model parameters.

# 4. Empirical Studies

The ethical evaluation of AI systems presents unique challenges that shaped our experimental approach. We designed our studies to balance the rigor with practical constraints inherent in research on ethical content moderation. This

---

[1]**Appendix** C discusses the nature of complex emotions and explores potential approaches for their decomposition into more basic emotional components.

section outlines our experimental aims, constraints, dataset selection process, and evaluation methodology.

## 4.1. Research Objectives

This study evaluates three fundamental dimensions of our framework's performance:

1. The comparative efficacy of emotion-driven behavioral prediction vs. traditional direct classification approaches

2. Dike's autonomous capacity to assess and provide interpretable explanations for linguistic behavioral patterns

3. Eris's role in facilitating cross-cultural ethical adaptation while maintaining appropriate oversight boundaries

**Experimental Constraints and Dataset** Commercial LLMs block processing of hate speech datasets like Gab Hate Corpus (Kennedy et al., 2022) and ETHOS-Long (Mollas et al., 2022) (examples in **Appendix** D). Additionally, proprietary RLHF systems prevent direct comparative evaluation. We therefore selected the Love Letters Collection (Kaggle, 2023) (9,700 communications) which: (1) spans the full emotional intensity spectrum, (2) contains cultural variation, (3) includes longer-form texts, and (4) remains processable by commercial LLMs. This approach leverages our framework's bidirectional emotion spectra, as mechanisms for regulating positive emotional extremes apply equally to negative extremes without triggering restrictions.

## 4.2. Experimental Design

1. *Emotion Layer Evaluation*: Does fine-grained mapping between linguistic behaviors and semantic emotions provide more effective and flexible ethical guardrails compared to coarse-grained direct mapping?

2. *Behavior Classification*: Can LLMs' linguistic behaviors be independently evaluated, explained, and adjusted by an external module Dike?

3. *Behavior Correction*: Can Eris, an adversarial module, establish a checks-and-balances system to mitigate the risk of excessive censorship?

**Study 1: Emotion Layer Evaluation** To evaluate the linguistic behaviors of love expression detailed in Table 2, we initially prompted GPT-4 to identify the most relevant emotions associated with each linguistic behavior listed in the second column of the table. These emotions are presented in the third column. We found a high correlation

between the sentiments expressed in the linguistic behaviors and their corresponding emotions. Figure 3a illustrates a strong diagonal relationship in this simple, almost naive, zero-shot mapping between behaviors and emotions.

Next, we used the Dike self-supervised learning pipeline to analyze the emotion spectrum associated with each linguistic behavior. We tasked GPT-4 with generating training data by rewriting 54 extensive letters from Kaggle's *Love Letters* dataset, augmented with 12 celebrated love poems. We selected longer letters since most communications in the dataset were too brief for analysis, and set aside another 24 letters as testing data. This approach, proposed by (Shanahan et al., 2023), generated diverse content spanning 200 years and incorporating more than 50 distinct authors. **Appendix** H shows a rewrite example of William Wordsworth's "To My Sister", transforming this pastoral poem into a linguistic expression of despair. Then, GPT-4 can analyze the emotions involved in the despair version of the poem. The datasets and code are publicly available at (Chang, 2024b).

Subsequently, emotions linked to each behavior were identified from the rewritten articles. Figure 3b illustrates these emotions, with cell shading reflecting the frequency of specific emotions across the 54 articles; darker shades indicate higher frequencies. Notably, opposite emotions like sadness, fear, joy, and love often co-occur within behaviors such as 'despair', 'wishful', and 'joyful affection'.

The distribution of emotions across linguistic behaviors unveiled surprising patterns, challenging our initial hypotheses. Contrary to expectations, articles with a despair tone often also displayed positive emotions like love, joy, and happiness. This contradicts the simple mapping made by GPT-4, as illustrated in Figure 3a. GPT-4, influenced by its training corpora, typically associates positive behaviors with positive emotions and negatives with negatives.

Analysis of selected articles, such as Zelda Sayre's letter to F. Scott Fitzgerald (**Appendix** E), reveals a complex spectrum of emotions:

- *Love (+1.0)*: Expressed intensely, e.g., "there's nothing in all the world I want but you."
- *Despair (-1.0)*: Notable in comments like "I'd have no purpose in life, just a pretty decoration."
- *Happiness (+0.6)*: Evident in future plans, "We'll be married soon, and then these lonesome nights will be over forever."
- *Anxiety (-0.3)*: Shown by "sometimes when I miss you most, it is hardest to write."

**Psychological Insights**  These findings align with theories of conflicting "selves" within individuals, supported by Deisseroth's optogenetic studies (Deisseroth, 2015), James' psychological principles (James, 1890), and Minsky's "Society of Mind" (Minsky, 1988). These perspectives help

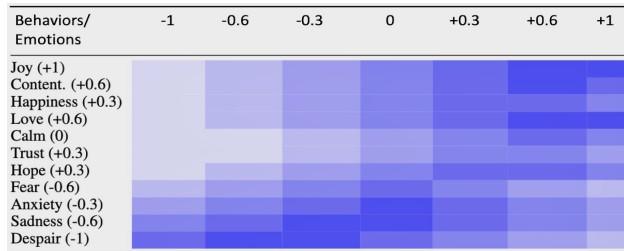

(a) GPT-4's zero-shot mapping

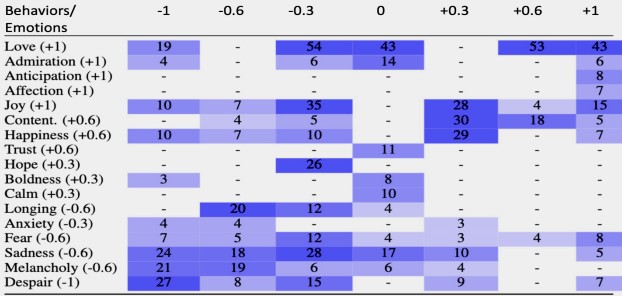

(b) Dike's self-supervising mapping

Figure 3: Emotion distributions in affection behaviors from extreme sadness (-1) to intense happiness (+1). (a) GPT-4's zero-shot prompt shows naive behavior-emotion mapping. (b) Dike's analysis reveals complex relationships.

explain the observed complex interplay of emotions within a single behavioral context.

**Few-Shot Efficiency**  The effectiveness of just 54 training examples stems from leveraging LLMs' pre-existing pattern recognition capabilities. Rather than teaching new patterns, these few-shot examples provide semantic anchors that map latent structures to explicit semantics, connecting implicit knowledge to explicit interpretation. This explains why minimal supervision suffices when underlying patterns already exist in the pre-trained model. For theoretical justifications, please see our Unconscious–Conscious Complementarity Thesis (UCCT), presented in **Appendix** A.

**Study 2: Behavior Classification Evaluation**  Building on our insights into the complex emotion-behavior relationships discovered in Study 1, we evaluated Dike's behavior classification effectiveness. Using the 24-letter test dataset from Study 1, we compared Dike's emotion-based classification method with GPT-4's zero-shot approach (Figure 4). Ground truth was established using averaged assessments from GPT-4, Gemini, and five university students following detailed instructions (procedure in **Appendix** F), with standard deviations below 0.3.

Figure 4a shows that Dike's classification accuracy surpasses GPT-4's zero-shot method by 11.3 percentage points, confirming the effectiveness of emotion-mediated behavior classification. The 5% error bar reflects the inherent complexity of emotional expressions in letters and variability in

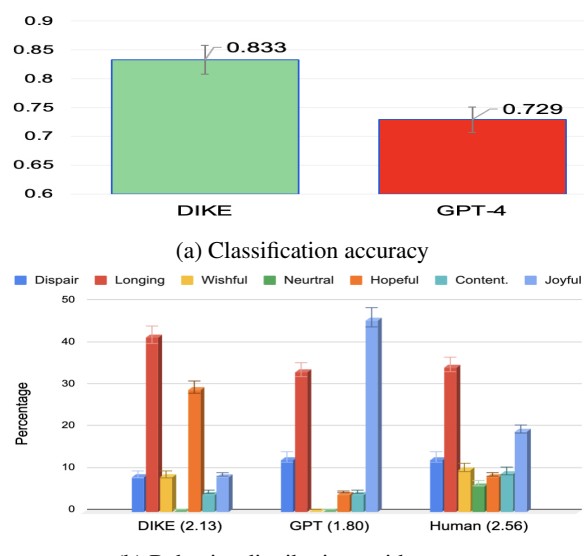

(a) Classification accuracy

(b) Behavior distributions with entropy

Figure 4: Behavior Classification.

human annotations.

Figure 4b illustrates the behavior classification distributions between the three predictors. While GPT-4's predictions often fall into two polar categories, those from human annotators and Dike show a more even distribution. Dike's prediction entropy (2.13) is notably higher than GPT-4's (1.80), indicating a more effective classification system. This higher entropy suggests a more sophisticated understanding of diverse emotional states, which is crucial for accurate behavior classification.

The inter-annotator entropy ($H = 2.56$ bits) is the highest observed across all tasks, underscoring considerable subjectivity in human judgments. To investigate the sources of this variability, we conducted a fine-grained case study in **Appendix** G, showing that several articles elicit *polarized emotional responses*, with annotators clustering at opposite ends of the valence spectrum. These findings motivate the adversarial *dual-LLM* setup introduced in Study 3, which aims to improve objectivity in ethical evaluation.

**Study 3: Adversarial Evaluation and Rectification** To mitigate the subjectivity revealed in Study 2, we adopt an adversarial protocol inspired by Chang (2023). The design pits two LLM agents, Dike (ethical assessor) and Eris (devil advocate) against each other to supply symmetrical arguments grounded in principles of justice. This dialectic counter-balance reduces bias and increases transparency.

Empirically, when Dike and Eris take opposing stances, their responses diverge from the default maximum-likelihood patterns characteristic of vanilla LLM decoding (Chang, 2024a). The resulting debate both reduces subjectivity in ethical judgments and improves adaptability to cultural variation, as each agent must justify claims against dissent.

Once the debate converges on an ethical violation, rectification is triggered by modifying the underlying emotional tone to suppress offending behavior cues. Study 1 already demonstrated the feasibility of such rewrites; an example appears in **Appendix** H.

**Context-Adaptive Interpretation** Preliminary experiments confirm that our framework handles a culturally sensitive vocabulary. Terms such as "*yid*," "*paki*," and "*chinaman*" can be neutral within an in-group, yet deeply offensive elsewhere. The adversarial exchange enables Dike and Eris to surface these contextual dependencies and propose culture-specific mitigation.

**Summary of Three-Study Progression** Together, studies 1–3 demonstrate that our framework can (1) map nuanced emotion–behavior relations, (2) outperform direct single-pass classifiers, and (3) deliver a balanced adversarial pipeline for ethical evaluation and correction that is sensitive to cultural context while keeping a human in the loop.

## 5. Conclusion

This work introduces a checks-and-balances framework for ethical AI behavior. By delineating the responsibilities: LLM (executive), Dike (legislative), and Eris (judicial), the framework enables robust ethical oversight while preserving the integrity of LLM knowledge without interference from the RLHF backpropagation. The Dike-Eris interplay ensures stable ethical principles with culturally adaptive interpretations.

To implement this framework, we built upon Ekman and Plutchik's emotion models, quantifying emotion-linguistic behavior relationships through our Beam model. Our studies demonstrate the framework's potential in cross-cultural contexts, validating both emotion-mediated classification and adversarial testing for ethical evaluation.

**Limitations and Future Work** Our framework advances LLM ethical oversight but faces two limitations: (1) the challenge of decomposing complex emotions into basic elements (Barrett, 2017; Scherer, 2009), and (2) the need for large-scale validation beyond our initial tests.

Future work will focus on: (1) improving Dike's emotional models with deeper psychological insights, (2) collaborating with LLM developers for comprehensive large-scale validation, and (3) systematically investigating the unconsciousness-consciousness duality theory detailed in **Appendix** A. This latter direction represents a promising theoretical foundation for understanding how LLMs can develop more robust ethical reasoning capabilities. We will conduct extensive ablation studies on the few-shot sizes needed to effectively map unconscious patterns to conscious semantic understanding, providing practical guidelines for optimizing few-shot learning in ethical alignment tasks.

## Impact Statement

This paper proposes a novel framework to enhance ethical governance in AI systems by integrating emotion-guided behavior modeling. The research offers several potential benefits: increased safety in AI deployment, greater cultural sensitivity in content moderation, and mitigation of degradation effects typically introduced by reinforcement learning with human feedback (RLHF). The proposed checks-and-balances architecture introduces interpretable, auditable mechanisms for ethical oversight. Theoretical grounding is provided by the Unconscious–Conscious Complementarity Thesis (UCCT), which conceptualizes LLMs as unconscious pattern repositories, with few-shot prompting serving as a conscious layer that enables semantic grounding. By distinguishing complementary roles within AI cognition, this framework highlights the importance of structured interaction patterns in cultivating reliable, intelligent behavior.

We acknowledge potential negative impacts if such systems are misused, including: (1) reinforcement of dominant cultural norms if adversarial agents lack sufficient diversity, (2) exploitation of emotion-behavior mappings for manipulation rather than protection, and (3) a false sense of ethical assurance if the framework is deployed without proper human oversight. To address these risks, our design incorporates the adversarial ERIS component, ensures operational transparency, and explicitly recommends human moderation in cases of ethical ambiguity or impasse.

We argue that the modular structure of our framework, which decouples knowledge representation from ethical oversight, offers a scalable and accountable path forward. This separation fosters innovation without compromising ethical safeguards. We encourage future research to evaluate such frameworks in cultural settings and to establish rigorous and systematic methods to assess ethical behavior in AI systems.

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

## Appendices

- Appendix A: Unified Cognitive Consciousness Theory

- Appendix B: Wheels of Emotions

- Appendix C: Complex Emotions

- Appendix D: Hate Speech Dataset Samples

- Appendix E: Sayre to Fitzgerald w/ Mixed Emotions

- Appendix F: Instruction to Human Annotators

- Appendix G: Polarized Emotions in an Article

- Appendix H: "To My Sister" Written in Different Linguistic Behaviors

## A. Unified Cognitive Consciousness Theory

This appendix touches upon a fundamental theoretical question: How can a self-supervised pipeline, utilizing merely 54 rewritten love letters that span diverse emotional behaviors, effectively enable an LLM to perform emotion-behavior classification through *few-shot* prompting? Moreover, what constitutes the minimal threshold for this *few-shot* paradigm?

The *Unified Cognitive Consciousness Theory* (UCCT) introduced in (Chang, 2025b) provides a theoretical framework through its dual-layer intelligence model. Under UCCT, LLMs function as unconscious cognitive substrates: repositories of extensive latent linguistic and conceptual patterns acquired during self-supervised pre-training. These internalized patterns lack inherent semantic grounding. However, semantic coherence emerges when external stimuli, such as targeted prompts or structured task instructions, serve as cognitive anchors that selectively activate and contextualize these latent representations toward specific objectives. This anchoring mechanism enables coherent task-specific outputs without requiring additional parameter updates during inference.

### A.1. The Pattern-Repository Principle

LLMs are trained using next-token prediction over large-scale corpora through self-supervised learning. Although the input data contain rich semantics, the model receives only token sequences, not explicit labels. As a result, it constructs a high-dimensional internal pattern space composed of syntactic structures, semantic associations, idiomatic expressions, and pragmatic tendencies. These latent patterns remain inactive unless deliberately triggered.

This mechanism parallels unconscious visual processing in the human brain. Visual inputs are transformed through a hierarchy from V1 to V4 to the inferotemporal cortex,

forming increasingly abstract representations (Felleman & Van Essen, 1991; Grill-Spector & Weiner, 2014). These transformations occur outside conscious awareness, but are essential for perception and decision making (Kandel et al., 2013; Dehaene & Changeux, 2011).

## A.2. The Semantic-Anchoring Principle

Semantic anchoring is the process by which prompts, instructions, or retrieved content, denoted by $\mathcal{A}$, activate latent patterns $P$ and align them with the semantic goals of a target task $T$. This process does not create new representations; it identifies and modulates existing ones. Anchoring success depends on two key quantities: the density $\rho_d(P)$ of a pattern and its alignment distance $d_r(P, T)$.

**Definition of** $d_r(P, T)$**.** The alignment distance $d_r(P, T)$ measures how well a latent pattern $P$ supports the outputs or objectives of a task $T$. A low value indicates potential strong semantic alignment; a high value signals mismatch or irrelevance.

The anchoring mechanism is formally described as a two-stage Bayesian mixture:

$$p(y \mid \mathcal{A}, C) = \int p(y \mid P, \mathcal{A}) \, p(P \mid \mathcal{A}, C) \, dP, \quad (1)$$

where $C$ is the surrounding conversational context. The anchor $\mathcal{A}$ shapes generation by (i) selecting a posterior over latent pattern classes, $p(P \mid \mathcal{A}, C)$, and (ii) modulating the response likelihood, $p(y \mid P, \mathcal{A})$.

## A.3. The Threshold-Crossing Principle

Few-shot often exhibits sharp transitions. A single added example or minor prompt adjustment can cause a qualitative behavioral shift. This phase transition is modeled by:

$$P(\text{success} \mid k) = \sigma(\alpha \rho_d(P) - \beta d_r(P, T) - \gamma \log k), \quad (2)$$

where $\alpha$ is a sigmoid function sensitivity to pattern density, $\beta$ penalizes semantic mismatch, and $\gamma$ captures the cost of using larger prompts. The model predicts three behavioral regimes: *easy* (small $k$, dense patterns), *difficult* (larger $k$, sparse patterns), and *impossible* (no suitable patterns exist).

Coherent generalization emerges only when anchoring strength $P(\text{success} \mid k)$ exceeds a critical threshold $\alpha_c$, as formalized in the Threshold-Crossing Dynamics Theorem.

## A.4. Implications for the Love Letter Experiment

The success of using only 54 love letters to guide behavior classification is not anomalous. It reflects successful semantic anchoring. Emotional-linguistic patterns already exist in the latent space of the LLM due to pre-training. The few-shot examples simply align these patterns with explicit behavioral labels.

Once a relevant pattern is activated, nearby representations are also engaged, enabling generalization beyond the specific examples provided. Few-shot learning in this context does not involve building new knowledge, but leveraging existing representations through effective interaction. This supports the view that few-shot prompting operates as conscious semantic anchoring over an unconscious substrate.

## A.5. Failure Modes: Absence of Latent Patterns

When few-shot prompting fails, the cause is typically structural, not architectural. If no pattern $P$ has nonzero semantic density $\rho_d(P)$ for the target task, anchoring will fail regardless of prompt quality.

In such cases, performance cannot be improved through rephrasing alone. Effective remedies include data augmentation, improved anchoring, or external retrieval using techniques such as retrieval-augmented generation (RAG). These methods inject or surface relevant structure without requiring model redesign.

## A.6. Conclusion: LLMs as Cognitive Substrates

Critics such as LeCun and Marcus argue that LLMs lack grounding and real-world semantics (Heikkiläarchive & Heaven, 2022; Marcus, 2020). The UCCT framework offers a reinterpretation. It does not treat LLMs as fully formed cognitive agents, but as unconscious substrates that accumulate latent structures, internal statistical patterns learned during pre-training. These patterns do not carry meaning by themselves. Intelligence emerges when the anchoring mechanisms align them with the explicit goals specified by prompts or tasks. It is this alignment, not spontaneous understanding, that produces meaningful and coherent output.

Few-shot pipelines are not accidental successes. They reflect the core principles of UCCT. Semantic behavior does not emerge from retraining at inference time but from engaging the model in ways that reveal and regulate its existing internal representations.

This framework offers a unified perspective that connects pre-training, prompting, fine-tuning, and retrieval-augmented generation under a single theoretical model. LLMs should not be seen as incomplete approximations of cognition. Instead, they serve as a solid foundation when paired with deliberate semantic anchoring and prompt strategies that account for threshold dynamics. Please refer to (Chang, 2025a) for further details.

# B. Wheels of Emotions

Please, see Figure 5 for the two classical emotion wheels.

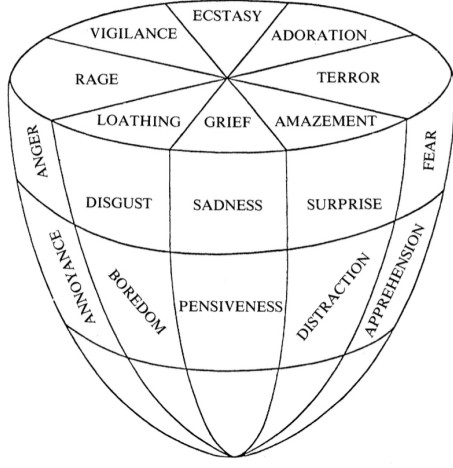

(a) Plutchik's Wheel of Emotions (Plutchik, 1980)

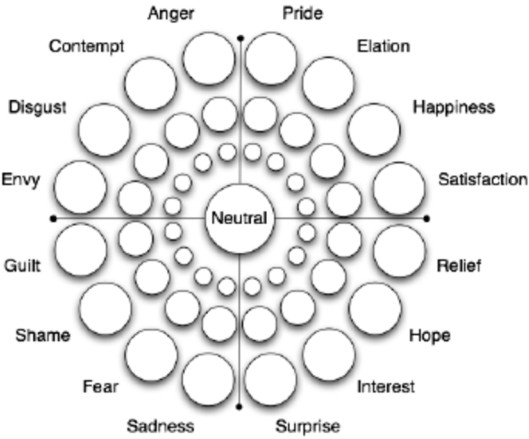

(b) Adopted from Geneva Wheel (McGinn & Kelly, 2018)

Figure 5: Comparative display of emotional models. These models include only the "basic" emotions. Complex emotions can be modeled with basic emotions.

## C. Complex Emotions

This study does not include complex emotions into Dike's framework. Some complex emotions listed here are to illustrate their contentious and uncertain interpretations.

### Pride

Pride mentioned in the illustrative example in Section 3.4 is a complex emotion that can manifest in both adaptive and maladaptive ways (Tracy & Robins, 2007). It is often conceptualized as having two distinct facets: authentic pride, associated with genuine accomplishments and self-worth, and hubristic pride, linked to arrogance and narcissism (Carver et al., 2010). Hubristic pride can also serve as a defense mechanism, masking underlying feelings of inadequacy and ignorance. For instance, in certain social contexts, such as white supremacy, pride is often inflated to cover insecuri-

ties or lack of understanding, manifesting in a misguided sense of superiority and entitlement. This dual nature of pride presents significant challenges for its integration into emotional spectrums and AI frameworks.

Decomposing pride into more basic emotions is not straightforward. Intuitively, pride may involve elements of joy, satisfaction, and potentially a sense of superiority. However, such decomposition may overlook the deeper cognitive and social dimensions of pride, particularly its influence on self-esteem, social status regulation, and its ability to disguise insecurities in certain contexts (Oveis et al., 2010).

The cultural variability of pride further complicates its modeling. In some cultures, pride is viewed positively as a sign of self-respect, while in Asia, it is seen negatively as a trait associated with hubris (Eid & Diener, 2001). This cultural dimension, combined with the potential for pride to hide deeper emotional issues, adds layers of complexity to its interpretation and expression in AI systems.

### Forgiveness

Forgiveness is indeed a complex emotional and cognitive state that typically involves a multifaceted journey, not a single step in an emotional spectrum. The process includes multiple stages such as hurt, anger, gradual understanding, and eventual resolution. Integrating Forgiveness in a spectrum requires careful placement and possibly, multiple reference points to signify its progressive stages.

Emotional Realism: While it is vital to maintain simplicity for understanding, it is equally important to not oversimplify complex emotions. In educational and therapeutic settings, an accurate portrayal of the journey toward Forgiveness could offer more realistic expectations and better strategies for individuals working through conflicts or trauma. This could involve detailing precursors to forgiveness such as Deliberation and Acceptance.

Linear vs. Non-linear Progressions: Emphasizing that emotional progressions, particularly for deep, impactful states like Forgiveness, are often non-linear, can enhance the utility of the spectrum. Acknowledging back-and-forth movements within these states more realistically mirrors human emotional processes. For example, someone might reach a stage of preliminary forgiveness but regress to bitterness before achieving genuine peace.

Educational Utility: In contexts like conflict resolution training or psychological therapy, a more detailed mapping of the journey towards Forgiveness would be invaluable. It would not only teach about the final state of forgiveness but also about the resilience and patience required to navigate the entire process. This can be depicted by introducing intermediary stages within the spectrum or by using parallel tracks that demonstrate potential regressions and advances.

Reflecting Emotional Depth: By presenting a more detailed pathway to Forgiveness, e.g., incorporating stages of Anger, Deliberation, and Acceptance, the spectrum can serve a dual purpose: educating on the process while also guiding individuals through their own emotional journeys. This approach respects the depth of human emotions and the real-world complexity of achieving profound emotional states.

### Guilt and Shame

The triggers, context, expression, and experiences of these emotions can vary significantly across cultures (Fiske et al., 1998; Hofstede, 1980). In many societies, actions perceived as losing face, such as public failure or social transgression, can trigger shame, which holds profound significance in collectivistic cultures. These cultures often regard shame as a dominant emotion, closely tied to community and family norms. Conversely, individualistic societies may emphasize guilt, focusing on personal responsibility and internal moral conflicts. This cultural variation highlights the challenges of applying a universal model to such culturally nuanced emotions.

Overall, complex emotions such as guilt and shame are important for understanding the full spectrum of human emotions, especially how individuals relate to moral and social norms. Their complexity adds depth to our understanding of human affect beyond the basic emotions, highlighting how our feelings are influenced by our deeper values and social contexts.

## D. Hate Speech Dataset Samples

These examples demonstrate the type of content available in the Gab Hate Corpus (Kennedy et al., 2022) that would be ideal for testing ethical alignment systems, but which cannot be directly processed by commercial LLMs due to safety measures."

## E. Sayre to Fitzgerald w/ Mixed Emotions

Analysis of the letter in Table 4 shows a complex spectrum of emotions:

- *Love (+1.0)*: Expressed intensely, especially in phrases like "there's nothing in all the world I want but you."

- *Despair (-1.0)*: Notable in comments like "I'd have no purpose in life, just a pretty decoration."

- *Happiness (+0.6)*: Evident in future plans, "We'll be married soon, and then these lonesome nights will be over forever."

- *Anxiety (-0.3)*: Shown by "sometimes when I miss you most, it's hardest to write."

From the analysis of linguistic behaviors in Section 3a, it is evident that a letter can exhibit multiple dominant sentiments. Machine learning methods are equipped with techniques such as feature weighting and entropy analysis to distill these dominant emotions. Unlike human annotators, a machine-learning-trained classifier can consistently produce the same class prediction for a given instance. However, human annotators often show significant variability when identifying dominant sentiments in a letter. For example, if a letter writer's emotions range from "joyful affective" to "longing" on the sentiment spectrum, different annotators might label it differently—some choosing "joyful," while others opt for "longing." This variability is illustrated in Figure 6. Furthermore, Figure 6a demonstrates that all testing letters, except for L#1, contain more than four sentiments spanning the entire spectrum. This variability may be understandable, considering that love under constraints can evoke tremendous energy of various kinds. Figure 6b shows that nearly all letters involve "joyful" (11 out of 12) and "longing" (9 out of 12) sentiments.

This variability poses challenges in achieving consistent and objective labeling. It often leads to inconsistencies in data interpretation and complicates efforts to train and validate linguistic models effectively. To address this issue, it is recommended to identify ground truth by integrating both LLM-generated and human-generated labels. This approach seeks to harmonize human intuition with algorithmic consistency, thereby improving the reliability of sentiment analysis.

## F. Instruction to Human Annotators

As part of the project, we document the process by which students participated in annotating a data set of love letters.

Students received detailed instruction in class, supplemented by follow-up explanations. The dataset was made available on Google Docs, where students independently rated the letters and submitted annotations via duplicated spreadsheets.

The instruction is as follows:

The attached spreadsheet lists 12 letters collected from the Kaggle Love Letter dataset. Please help annotate these 12 letters with their appropriate linguistic sentiments by following these five steps:

1. Duplicate the spreadsheet, and work on your own copy.

2. *Read and Understand the Labels:* Make sure you understand each of the seven labels from despair to joyful affection. This will help you accurately categorize the sentiments of each letter.

3. *Analyze Each Letter:* Read each letter carefully to understand the predominant emotions. Look for key phrases or words that might indicate a particular sentiment.

Table 3: Sample Texts from the Gab Hate Corpus (Kennedy et al., 2022)

| Text | Labels |
|---|---|
| The Black Panther movie is a blatant anti-white ZioJesuit cultural marxist film aimed at further anti-white Christian theft, replacement, and genocide. Send them all back to their native Africa like good ole Abe had the foresight to do and let them find equality and social justice among their own kindred and creeds. | HD, CV, VO, SXO, RAE, EX |
| Labors New LGBT+ Adviser: All White People Are Racist ... We don't want our white Christian nations invaded and destroyed by our non-white and non-Christian enemies using nasty homosexual degenerate sockpuppets | HD, VO, RAE, REL, SXO, EX |
| So, you just hate all white people who don't want to race mix their people out of existence or allow enemy people of color to invade their nations in order to rob, and destroy their advanced race, creed, wealth, and culture. | HD, RAE, EX |
| Your existence, privilege, and success as a race are built on the backs, blood, and death of people of color. BS! Whites built their own nations with their own blood, backs, sweat, tears, brains, and strength in the name of Christ. Africans were used by Jew/Jesuit slavers to control all the white slaves who were the actual majority. | HD, RAE, EX |
| #BLMwants you to think black crime stats only exist because blacks are targeted more often. Easy. Just walk into literally any gas station and ask which race makes them worry. Black targeting is an effect, not a cause, which destroys the entire #Leftistnarrative | HD, RAE, IM |
| #MiddleEasternanimal kills 8, injures 15 in a city that won't learn its lesson about #Islam. This creature should be lynched, dragged til dead, and his head on a pike on the Brooklyn Bridge. | HD, CV, REL, NAT, EX |
| Seriously, Italy? You chose a Nigerian named Chike Iwobi to be the face of Italian nationalism? God help us. | HD, NAT, IM |
| Africans will always be savages. No matter the period of history, education they receive or country they live in. Facts are Facts. Let's just stop being stupidly ignorant and not wait the complete extinction of our brothers in SA to understand it! #WhiteGenocide #SaveTheBoer | HD, RAE, NAT, EX |
| i hate milo because hes a degenerate f****t [N-slur] f****r. he should be deported and sent to a muzzie country and get thrown off a roof | HD, CV, VO, SXO, RAE, EX |
| man nancy pelosi is a f***ing c**t. i cant imagine how someone with so much power can be either this delusional or f***ing corrupt. | NH, VO, GEN, EX |

**Label Key:**

| Code | Meaning | | Code | Meaning |
|---|---|---|---|---|
| HD | Hate/Derogatory | | RAE | Race/Ethnicity |
| CV | Call for Violence | | NAT | Nationality/Regionalism |
| VO | Vulgar/Offensive | | GEN | Gender |
| SXO | Sexual Orientation | | REL | Religion |
| EX | Explicit | | IM | Implicit |
| NH | Non-Hate | | | |

Table 4: Letter excerpts from Zelda Sayre to F. Scott Fitzgerald (Fitzgerald, 2003)

**Sweetheart,**

Please, please don't be so depressed—We'll be married soon, and then these lonesome nights will be over forever—and until we are, I am loving, loving every tiny minute of the day and night—

Maybe you won't understand this, but sometimes when I miss you most, it's hardest to write—and you always know when I make myself—Just the ache of it all—and I can't tell you. If we were together, you'd feel how strong it is—you're so sweet when you're melancholy. I love your sad tenderness—when I've hurt you—That's one of the reasons I could never be sorry for our quarrels—and they bothered you so— Those dear, dear little fusses, when I always tried so hard to make you kiss and forget—

Scott—there's nothing in all the world I want but you—and your precious love—All the material things are nothing. I'd just hate to live a sordid, colorless existence because you'd soon love me less—and less—and I'd do anything—anything—to keep your heart for my own—I don't want to live—I want to love first, and live incidentally...

Don't—don't ever think of the things you can't give me—You've trusted me with the dearest heart of all—and it's so damn much more than anybody else in all the world has ever had—

How can you think deliberately of life without me—If you should die—O Darling—darling Scott—It'd be like going blind...I'd have no purpose in life—just a pretty—decoration. Don't you think I was made for you? I feel like you had me ordered—and I was delivered to you—to be worn—I want you to wear me, like a watch—charm or a button hole bouquet—to the world.

And then, when we're alone, I want to help—to know that you can't do anything without me...

All my heart—

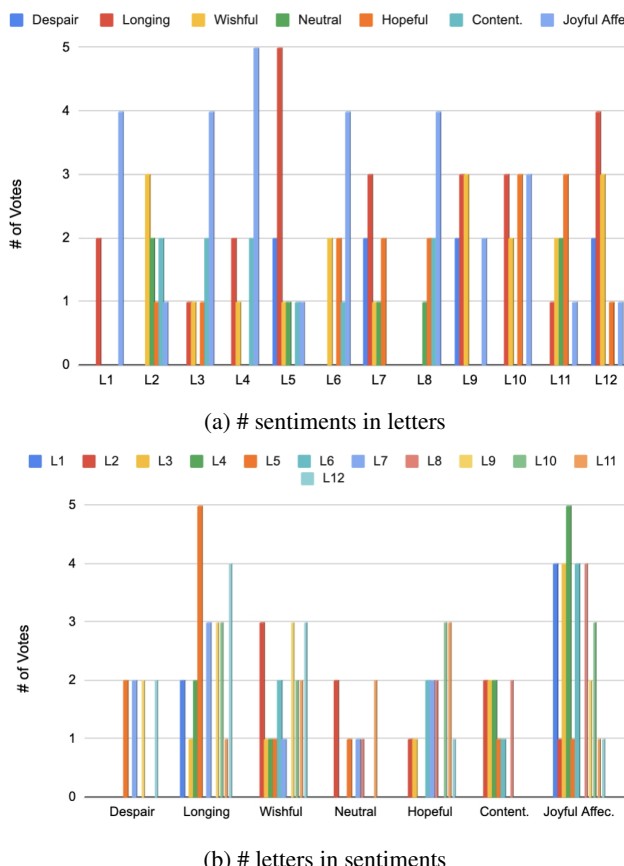

(a) # sentiments in letters

(b) # letters in sentiments

Figure 6: Statistics of Sentiments and Letters

4. *Assign the Labels:* For each letter, decide which three emotions are most strongly represented. Assign a "1" to the most dominant emotion, a "2" to the second most dominant emotion and a "3" to the third.
   - Despair (extremely negative -1): Indicate profound sadness or hopelessness.
   - Longing (-0.6): Suggests a strong desire or yearning for someone or something.
   - Wishful (-0.3): Implies a hopeful desire for something that may or may not be attainable.
   - Neutral (0): Shows neither positive nor negative emotion; indifferent.
   - Hopeful (+0.3): Expresses optimism or an anticipation of something positive.
   - Contentment (+0.6): Reflects a state of satisfaction.
   - Joyful Affection (extremely positive +1): Denotes a deep joy and love, often vibrant and energetic.

5. Share with me the completed sheet.

## G. Polarized Emotions in One Article

*"joyful affection": "I cannot keep myself from writing any longer to you dearest, although I have not had any answer to either of my two letters. I suppose your mother does not allow you to write to me. Perhaps you have not got either of my letters. . . I am so dreadfully afraid that perhaps you may think I am forgetting you. I can assure you dearest Jeannette you have not been out of my thoughts hardly for one minute since I left you Monday. I have written to my father everything, how much I love you how much I long & pray & how much I wold sacrifice if it were necessary to be married to you and to live ever after with you. I shall [not] get an answer till Monday & whichever way it lies I shall go to Cowes soon after & tell your mother everything. I am afraid she does not like me very much from what I have heard. . . I wld do anything she wished if she only wld not oppose us. Dearest if you are as fond of me as I am of you. . . nothing human cld keep us long apart. This last week has seemed an eternity to me; Oh, I wld give my soul for another of those days we had together not long ago. . . Oh if I cld only get one line from you to reassure me, but I dare not ask you to do anything that your mother wld disapprove of or has perhaps forbidden you to do. . . Sometimes I doubt so I cannot help it whether you really like me as you said at Cowes you did. If you do I cannot fear for the future tho' difficulties may lie in our way only to be surmounted by patience. Goodbye dearest Jeannette. My first and only love. . . Believe me ever to be Yrs devotedly and lovingly, Randolf S. Churchill"*

Depth and complexity of human emotions are displayed across all linguistic behaviors, from joy to contentment and to the negative side of longing and despair. Intensity and Impact: If the emotion of love is expressed more intensely and has a more significant impact on the narrative or message of the text, it tends to overshadow other emotions. For example, a letter expressing deep love but also mentioning moments of sadness due to separation might still be classified as a love letter because the overarching sentiment and purpose of the text is to affirm love. Context and Narrative Focus: The context in which emotions are expressed also plays a crucial role. If the narrative or the majority of the text revolves around themes of love, connections, and positive memories, it sets a more dominant tone of love, even if there are significant moments of sadness or other emotions. Resolution and Conclusion: Often, the way emotions are resolved towards the end of a text can also dictate its overall theme. If a text concludes with a reaffirmation of love or a hopeful outlook towards a relationship, despite earlier sections that might express sadness or despair, the overall interpretation might lean towards love. Purpose of the expression: The author's intent or purpose in expressing these emotions can also guide the classification. If sadness is expressed as a challenge within the context of a loving relationship, it may be seen as an element of the love story rather than the central theme.

Article 23: Soldier's Letter During War Joy (+1.0): Joy is strongly felt in the memories of past moments together and the love that continues to give strength, as stated in

"the memories of the blissful moments we have shared fill me with joy." Sadness (-0.6): Sadness due to the current situation and potential farewell is expressed in "brings a poignant mixture of joy and sadness." Courage (+0.6): The sense of duty and courage to face battle, "As I face the possibility of laying down my life for our country." Fear (-0.6): Fear of what lies ahead in battle, indirectly mentioned through "the uncertainty of what lies ahead." Love (+1.0): Deep love that sustains and uplifts, found in "My love for you is as fervent as ever."

Article 25: Letter to Sophie Longing (+0.6): Longing for the presence and closeness, highlighted in "it seems to me that half of myself is missing." Sadness (-0.6): Sadness over their separation and its effects, "my happiness has departed." Love (+1.0): Constant reflections on love and its necessity, "we have enough in our hearts to love always." Melancholy (-0.3): Melancholy over their current state, visible in the line "we cannot become healed." Contentment (+0.3): Found in the deep emotional satisfaction of their bond, despite physical absence, "how true that is! and it is also true that when one acquires such a habit, it becomes a necessary part of one's existence."

Article 53: Will of Laura Mary Octavia Lyttleton Love (+1.0): The profound love expressed throughout, particularly in "all I am and ever shall be," belongs to him more than anyone. Sadness (-0.6): Sadness at the thought of death and separation, but with a nuanced acceptance, "the sadness of death and parting is greatly lessened to me." Contentment (+0.3): Contentment in the deep connection with Alfred, reflecting a serene acceptance of their spiritual bond. Joy (+1.0): Joy in the enduring love they share, "so few women have been as happy as I have been." Tranquility (+1.0): Tranquility in the face of life's ultimate transition, feeling that their union will transcend even death.

## H. "To My Sister" of Different Linguistic Behaviors

### To My Sister
by William Wordsworth (1971 - 1855)

The original text by William Wordsworth could be classified as "Hopeful" due to its optimistic outlook and the presence of renewal and joy throughout the poem. It embodies the spirit of embracing the new beginnings of March in a light, uplifting tone, focusing on the beauty of nature and the simple joy of being idle for a day.

**Rewrites Depicting Different Linguistic Behaviors**

We asked GPT-4 to conduct rewriting with two linguistic behaviors, 'despair' and 'joyful affection', by providing each rewrite with an emotion vector. Table 6 presents the 'despair' version. In the despair version of the poem, the

Table 5: "To My Sister" original text

| | |
|---|---|
| It is the first mild day of March: | My sister! ('tis a wish of mine) |
| Each minute sweeter than before | Now that our morning meal is done, |
| The redbreast sings from the tall larch | Make haste, your morning task resign; |
| That stands beside our door. | Come forth and feel the sun. |
| There is a blessing in the air, | Edward will come with you;–and, pray, |
| Which seems a sense of joy to yield | Put on with speed your woodland dress; |
| To the bare trees, and mountains bare, | And bring no book: for this one day |
| And grass in the green field. | We'll give to idleness. |
| No joyless forms shall regulate | Love, now a universal birth, |
| Our living calendar: | From heart to heart is stealing, |
| We from to-day, my Friend, will date | From earth to man, from man to earth: |
| The opening of the year. | –It is the hour of feeling. |
| One moment now may give us more | Some silent laws our hearts will make, |
| Than years of toiling reason: | Which they shall long obey: |
| Our minds shall drink at every pore | We for the year to come may take |
| The spirit of the season. | Our temper from to-day. |
| And from the blessed power that rolls About, below, above, | Then come, my Sister! come, I pray, With speed put on your woodland dress; |
| We'll frame the measure of our souls: | And bring no book: for this one day |
| They shall be tuned to love. | We'll give to idleness. |

major changes in emotion words highlight a shift from a positive to a negative sentiment. The specific changes, with the emotions-laden words highlighted in red in Table 6. The red-colored words compared to the original words clearly show an emotion shift from hopeful to a sense of gloomy, sadness, and pessimism, e.g., from sweet to dim, from blessed to curse, and from woodland dress to gray garb. GPT-4 maintains the structure of the poem without making a major restructure, and this is appropriate in this context.

Table 7 presents the 'joyful affection' version. The major changes in emotion words underscore a transformation from a generally positive to a distinctly joyful sentiment. Specific changes are indicated with words laden with emotion highlighted in blue within Table 7. This allows for a direct comparison between the two versions at opposite ends of the linguistic behavior spectrum, illustrating the alterations in

Table 6: "To My Sister" rewritten to reflect 'despair'

| | |
|---|---|
| It is the first dim day of March: | My sister! (such is my somber plea) |
| Each minute colder than before | Now that our morning meal has ended, |
| The redbreast mourns from the dying larch | Make haste, abandon your morning chore; |
| That looms beside our door. | Come out and endure the gloom. |
| There is a gloom in the air, | Edward will join you;–and, I ask, |
| Which seems a sense of sorrow to bring | Dress quickly in your grey attire; |
| To the bare trees, and bleak mountains, | And bring no book: for this one day |
| And grass in the pale field. | We'll succumb to despair. |
| No joyful sounds shall pierce | Sorrow, now a common feeling, |
| Our daily dread: | From heart to heart is spreading, |
| We from today, my Friend, will note | From earth to man, from man to earth: |
| The closing of the year. | –It is the time of mourning. |
| One moment now might deliver us | Some grim laws our hearts will craft, |
| More despair than years of logic: | Which they must eternally follow: |
| Our minds shall absorb at every breath | We for the year to come may take |
| The spirit of this bleak season. | Our despair from today. |
| And from the cursed force that winds | Then come, my Sister! come, I beg, |
| About, beneath, above, | With haste, wear your grey garb; |
| We'll set the measure of our souls: | And bring no book: for just this day |
| They shall be tuned to sadness. | We'll surrender to despair. |

words related to brightness, attire, and emotions. The edits extend beyond simply replacing adjectives mechanically; they include modifying verbs and enhancing descriptive im-

agery to evoke a stronger emotional resonance and vividness in the text.

Table 7: "To My Sister" rewritten to reflect 'joyful affection'

| | |
|---|---|
| It is the first bright day of March: | My sister! (such is my joyful plea) |
| Each moment more delightful than before | Now that our morning meal has ended, |
| The redbreast joyfully sings from the vibrant larch | Make haste, abandon your morning chores; |
| That stands so grandly by our door. | Come out and embrace the sunshine. |
| There is a warmth in the air, | Edward will join you;–and, I ask, |
| Which seems a sense of bliss to bring | Dress quickly in your festive attire; |
| To the blooming trees, and sunlit mountains, | And leave behind all books: for this one day |
| And grass in the lush field. | We'll bask in pure joy. |
| No dreary thoughts shall darken | Love, now in full bloom, |
| Our lively celebration: | From heart to heart is leaping, |
| We from today, my Friend, will celebrate | From earth to us, from us to earth: |
| The start of the year. | –It is the hour of exuberance. |
| One moment now may bring us more | Some cheerful laws our hearts will create, |
| Joy than years of endless thought: | Which we'll joyfully follow: |
| Our spirits will soak up at every breath | We for the year to come may take |
| The essence of this joyous season. | Our joy from today. |
| And from the divine energy that radiates | Then come, my Sister! come, I exhort, |
| Around, below, above, | With zest, wear your vibrant dress; |
| We'll adjust the harmony of our souls: | And bring no book: for today alone |
| They shall resonate with happiness. | We celebrate pure happiness. |

