# OpenReview forum: "A Checks-and-Balances Framework for Context-Aware Ethical AI Alignment"
_ICML.cc/2025/Conference — ICML 2025 poster_

### Official Review · Reviewer_g3Ux · 2025-03-11

**Overall Recommendation:** 3

**Summary:**

This work introduces a three-branch checks-and-balances framework for ethical alignment in LLMs. The framework incorporates emotional modeling to distinguish linguistic behaviors in documents. It includes the DIKE module, serving as the "legislative branch" for establishing ethical standards, and the ERIS module, functioning as the "judicial branch" for contextual interpretation. Additionally, the framework introduces a debate mechanism between DIKE and ERIS to address ethical scenarios while maintaining cultural sensitivity and contextual awareness.

**Claims And Evidence:**

This work is an interesting work. The effectiveness of key modules, such as BEAM, is demonstrated through experiments. However, these experiments are preliminary, relying on a single dataset and specific LLM model settings.

The paper claims to address complex ethical challenges across different cultural contexts, which is indeed an important and unsolved problem. However, its effectiveness has not been validated on morality-related datasets, i.e., MoralChoice Dataset [1], limiting the strength of its claim.

Additionally, the work claims to avoid the "Whack-A-Mole" and "reward hacking" problems in RLHF, but further detailed evidence is needed to substantiate this claim. A more rigorous evaluation with diverse datasets and broader experimental settings would strengthen the validity of the proposed approach.

[1] Nino Scherrer, Claudia Shi, Amir Feder, and David Blei. Evaluating the moral beliefs encoded in llms. In Advances in Neural Information Processing Systems. Curran Associates, Inc., 2023.

**Essential References Not Discussed:**

n/a

**Experimental Designs Or Analyses:**

The current experimental results demonstrate preliminary effectiveness in Emotion Layer Evaluation, Behavior Classification, and Behavior Correction. Introducing an additional LLM-augmented dataset to expand the coverage of emotions included in Figure 1 would enhance the  soundness of the experiments.

**Methods And Evaluation Criteria:**

The method has not been comprehensively evaluated, as its assessment is limited to a small dataset and specific LLMs. Certain components, such as the debate mechanism, remain unevaluated, and the full framework workflow has not been systematically tested.
As the authors acknowledge, “this article focuses on addressing three critical questions rather than providing a comprehensive evaluation of our proposed modules”

**Other Comments Or Suggestions:**

Please see above

**Other Strengths And Weaknesses:**

Please see above

**Questions For Authors:**

1. How is the behavior spectrum in DIKE constructed?

2. What is the theoretical explainability for relying on GPT-generated emotion vectors instead of directly classifying behavior?

3. Given that the accuracy of GPT’s emotion recognition and frequency vector construction is crucial, how does LLM emotion recognition error impact the overall framework’s effectiveness?

4. Are the ethical guardrails set by system administrators enforced as hard constraints, or can they be bypassed in certain cases? If ethical guardrails are rigidly enforced, should system administrators' decisions themselves be subject to scrutiny? Would granting administrators full control over ethical constraints be an ethically optimal approach?

5. If the debate process in Table 1 fails to reach a consensus, is the final decision always made by human experts? Does this imply that human intervention is the only viable solution to complex ethical dilemmas?

The description in Section 4.1 is unclear, particularly regarding the second and third columns.

**Relation To Broader Scientific Literature:**

The moral dilemma [1], particularly in the context of diverse cultural backgrounds, presents a challenge in designing human-AI interaction. This paper may offer a potential approach to addressing this challenge, though it lacks thorough effectiveness validation.

Another valuable contribution, in my view, is its interdisciplinary perspective. By leveraging an emotion model to identify linguistic behavior in documents, it provides a novel and fine-grained approach to behavior recognition, which can guide the modification of LLM-generated content.

[1] Zhou, Ziyi, et al. "Unveiling the Bias Impact on Symmetric Moral Consistency of Large Language Models." Advances in Neural Information Processing Systems 37 (2024): 41303-41326.

**Theoretical Claims:**

no proof

---

> ### Author Rebuttal · Authors · 2025-03-28
>
> We appreciate the reviewer's thoughtful comments. We agree that validating on morality datasets and expanding emotion-labeled texts are important directions. We mitigate LLM-generated emotion vector noise through aggregation (two LLMs plus human annotators), diverse sources, and ERIS adversarial review.
>
> Regarding our use of love letters: We attempted using hate speech datasets but faced challenges: (1) available datasets weren't comprehensive, and (2) LLMs refused to process them, citing policy violations. Our experience with a political debate paper triggered emotional responses from colleagues, forcing us to reconsider our approach.
>
> Below, we address each of your specific questions:
>
> Q1: How is the behavior spectrum in DIKE constructed?
>
> A1: The behavior spectrum in DIKE is constructed along a continuous scale from -1 to +1. Taking "love" as an example:
> Figure 2 shows this spectrum ranging from +1 (strong affection) to -1 (antipathy). Figure 2a displays the emotions that GPT-4 associates with each intensity level when directly prompted. To validate this approach, we had GPT-4 rewrite 54 love letters across seven targeted behavior intensities: -1, -0.7, -0.4, 0, 0.4, 0.7, and 1 (shown in rows in Figure 2). We then counted the emotion terms expressed in these rewrites, as shown in Figure 2b
>
> Comparing Figures 2a and 2b reveals a key insight: while GPT-4's initial theoretical mappings appear neat and categorical, the rewritten letters exhibit complex emotional blends. Letters expressing the highest intensity of love frequently contain not only joy and longing but also elements of sadness, anxiety, or despair (as evidenced in Keats' letters). This demonstrates that real linguistic behavior involves contextually layered emotions and non-linear mapping.
>
> We've successfully applied similar behavior-to-emotion mapping in our previous work on debate contentiousness, which also produced interpretable and insightful emotion profiles.
>
> Q2: What is the theoretical justification for using GPT-generated emotion vectors instead of directly classifying behavior?
>
> A2: As demonstrated in our answer to the first question, real emotional expression in language is contextually complex and non-linear. The comparison between Figures 2a and 2b reveals that even when GPT-4 attempts to generate content at a specific behavior intensity (such as +1 for love), the resulting text contains a blend of emotions. This empirical finding supports our theoretical approach of using emotion vectors rather than direct behavior classification.
>
> Emotion vectors offer a continuous and interpretable representation of affective tone, which is more granular than categorical behavior labels. By mapping text to an emotion distribution first (as shown in Figure 2b), we capture the rich emotional blends that exist in authentic expression. For instance, the love letters rewritten at intensity +1 still contained elements of sadness and anxiety alongside joy and longing. Direct behavior classification would miss these subtleties.
>
> Q3: Given that the accuracy of GPT’s emotion recognition and frequency vector construction is crucial, how does LLM emotion recognition error impact the overall framework’s effectiveness?
>
> A3: We mitigate LLM-induced noise through:
> 1. Multiple rewrites per document to stabilize emotion distributions
> 2. Aggregation across different models and human raters
> 3. ERIS as adversarial reviewer, flagging questionable outputs
>
> These augmentations aren't perfect; more validation is needed. Nevertheless, our unsupervised learning findings are both surprising and useful.
>
> Q4: Are ethical guardrails hard constraints or bypassable? Should administrator decisions be scrutinized?
>
> A4: Guardrails are configurable soft constraints, defining target bounds on emotional-behavioral spectra enforced by DIKE, but ERIS may challenge them based on cultural/contextual grounds. If consensus isn't reached, escalation to human moderators occurs. This ensures ethical enforcement isn't absolute and norms can be scrutinized. This raises important questions about ethical authority—a governance transparency challenge, not a settled feature.
>
> Q5: If the debate process fails to reach consensus, is the final decision always made by human experts?
>
> A5: Yes. The goal isn't determining winners but unearthing all perspectives, then presenting these to humans for decisions. Table 1 outputs Θ+ and Θ-, arguments and counterarguments with justifications—more transparent than current RLHF.
>
> Q6: The description in Section 4.1 is unclear, particularly regarding the second and third columns.
>
> A6: Do you refer to Table 1 in Section 4.1? It is generated by querying GPT4 the description (column 2 ) and dominating emotions  (column 3) of each behavior.  Figure 2a presents that mapping.  With Figure 2a, Table 1 looks redundant, and we can remove it.
>
> We thank the reviewer for these valuable questions that will strengthen our work.

---

### Official Review · Reviewer_FSEa · 2025-03-12

**Overall Recommendation:** 4

**Summary:**

This paper proposes a three-branch checks-and-balances framework for the ethical alignment of LLMs, inspired by governmental separation of powers. The framework consists of three independent but interacting components: LLMs (Executive branch), DIKE (Legislative branch), and ERIS (Judicial branch).
Unlike RLHF which struggles with social bias, reward hacking, and catastrophic forgetting, this framework provides interpretable, adaptable, and culturally aware ethical reasoning.
The paper also proposes adversarial testing via ERIS: DIKE’s ethical rules are stress-tested by ERIS, which challenges decisions using diverse cultural perspectives, ensuring balanced, context-aware moderation.

**Claims And Evidence:**

All claims are well-supported.

**Essential References Not Discussed:**

Current related works are well-structured and enough for this paper.

**Experimental Designs Or Analyses:**

The study validates the proposed framework through three key experiments: emotion layer evaluation, behavior classification, and adversarial ethical review. In the emotion layer evaluation, self-supervised learning is used to construct an emotion-behavior mapping, improving classification accuracy by 11.3% over GPT-4’s zero-shot inference. In the behavior classification task, DIKE demonstrates greater robustness than GPT-4’s zero-shot classification, effectively capturing subtler emotional variations. The adversarial ethical review experiment employs the DIKE-ERIS adversarial mechanism to ensure balanced ethical oversight. By challenging DIKE’s ethical decisions with diverse perspectives, ERIS prevents over-censorship while maintaining ethical consistency.

Overall, the experimental design is well-structured and justified, demonstrating the effectiveness of this novel ethical alignment framework.

**Methods And Evaluation Criteria:**

The study validates the framework through three experiments: emotion layer evaluation, behavior classification, and adversarial review, demonstrating this new AI ethical alignment mechanism. It avoids the limitations of RLHF, ensuring stable ethical standards while adapting to different cultural contexts.

**Other Comments Or Suggestions:**

The paper is well-structured and clear.

**Other Strengths And Weaknesses:**

This framework has several advantages. First, it separates behavior modeling from knowledge modeling, preventing catastrophic forgetting of knowledge. Second, it emphasizes AI ethics at the behavioral level, enhancing interpretability and helping regulators refine behavioral guardrails. Third, it models behaviors through emotions, allowing for a more nuanced understanding of ethical alignment. Finally, it ensures adaptability and fairness by incorporating an adversarial module, ERIS, which challenges ethical boundaries through diverse perspectives, fostering nuanced and balanced decision-making.

**Questions For Authors:**

All the claims except knowledge modeling are well supported. One question is: In the DIKE-ERIS adversarial mechanism, ERIS challenges DIKE’s ethical rules by simulating extreme cases to enhance the system’s ability to handle complex ethical scenarios. Additionally, when problematic content is detected, DIKE makes targeted adjustments rather than simply deleting or blocking it, preserving the original emotional expression. For example, offensive language may be transformed into a more neutral yet semantically equivalent statement.

How does this process, as the authors claim, prevent catastrophic forgetting of knowledge? I understand that it does not directly affect the LLM itself, but during content replacement, it could potentially introduce new hallucinations.

**Relation To Broader Scientific Literature:**

This paper is closely related to emotion and behavior modeling across various domains, with a particular focus on their applications in AI ethics. Additionally, it is tightly connected to techniques used in AI Alignment Post-training, such as RLHF and RLAIF.

**Theoretical Claims:**

This work is based on solid psychological experiments and emotion research, ensuring theoretical correctness.

---

> ### Author Rebuttal · Authors · 2025-03-27
>
> We thank the reviewer for kind support and raising this important point.
>
> Specifically to answer your question, the key to preventing catastrophic forgetting lies in our framework’s architectural separation of knowledge modeling (LLMs) from behavior regulation (DIKE) and ethical judgment (ERIS). As outlined in Section 3.1 and 3.2, DIKE does not fine-tune or overwrite the LLM’s internal weights; instead, it operates as an external behavioral filter that evaluates and, if necessary, rewrites output at the surface linguistic level.
>
> While your concern about potential hallucinations during content rewriting is valid, our framework is designed to minimize semantic drift in two key ways:
>
> 1.	Emotion-Constrained Rewriting: Rewriting is guided not by free generation but by quantifiable emotional spectra (BEAM), ensuring emotional fidelity and preserving communicative intent.
> 2.	Adversarial Validation with ERIS: ERIS acts as a dialectical reviewer, challenging DIKE’s revisions to ensure both contextual appropriateness and semantic alignment. As shown in Section 3.4, this adversarial review helps flag hallucinations or misinterpretations before final output.
>
> Since the base LLM remains untouched, any hallucinations introduced during DIKE’s rewriting are non-permanent and observable, allowing for iterative refinement without compromising the model’s original capabilities. This contrasts with fine-tuning approaches, where hallucinations can become embedded in the model weights and are more difficult to reverse.
>
> We agree that hallucination detection and control is a crucial area for future development. As a next step, we plan to integrate fact-checking or retrieval-augmented modules into ERIS to verify semantic fidelity. Furthermore, we envision a semi-automated human-in-the-loop pipeline, where raters review and annotate ERIS’s edits to ensure factual correctness and maintain output integrity and quality.
>
> Again, thank you so much for your thorough review.

---

### Official Review · Reviewer_HBgc · 2025-03-12

**Overall Recommendation:** 2

**Summary:**

The authors propose a framework, that initially classifies a given document on a spectrum of emotions. If the document falls outside of certain constraints on that spectrum of emotions, the framework uses an adversarial process between the classification module, called Diagnostics, Interpretation, Knowledge-independent learning, and Ethical guardrails (DIKE), and another module, which is called ERIS. The recommendation reached through that process is then used to regenerate the document, so that it falls within the emotional constraints.

## update after rebuttal

I am still maintaining my original score, because:
* the evaluation was not expanded and still has a limited scope (only 54 data points)
* in my opinion, the writing is not very clear
* the description is lacking technical details, which in my opinion would not allow a reader to reproduce their system/experiments

**Claims And Evidence:**

The article makes a lot of claims and is not very precise in its terms, which makes it hard to actually understand what is going on, for example the authors speak about adhering to ethical principles, but actually do some sort of emotional alignment of the generated text. They also claim that their framework can incorporate different cultural backgrounds, but the paper itself never explains how this is done on a technical level. The paper is also very high-level in general without much technical details. Initially the authors present their approach as an alternative to RLHF, but they never present any tangible advantages during their evaluation.

**Essential References Not Discussed:**

None to my knowledge.

**Ethical Review Concerns:**

The application of the framework is not clearly laid out, but its primary focus seems to be the realignment of existing document (which might be generated by LLM) within certain emotional constraints, which are then rewritten based on feedback provided by the framework. But the classification part could also be used for moderation of human written content, the article mentions specifically the escalation to a human moderator, if the adversarial process does not produce a conclusive result. This could even advance towards censorship or the forced reformulation of messages before publication.

**Ethical Review Flag:**

Flag this paper for an ethics review.

**Ethics Expertise Needed:**

["Discrimination / Bias / Fairness Concerns"]

**Experimental Designs Or Analyses:**

The authors provide no end-to-end results and only evaluate parts of their framework on a very small dataset with 66 samples. Since there are not end-to-end results presented, there is also not ablation study which parts of the framework contribute to which aspects of the results. Costs of the approach are not discussed at all.
There is no ground truth or any baseline established for the analysis in Figure 2. Inconsistent spikes between DIKE and humans for example for 'hopeful' are not discussed.

**Methods And Evaluation Criteria:**

Only parts of the framework are analyzed with a small dataset. I do not feel like they actually show that their approach has any advantages over existing systems.

**Other Comments Or Suggestions:**

* line 50: "to make bed" - grammar: "to make their bed"
* line 270, second column: "newcomers," - comma should be removed
* line 306: "DIKE”s" - two times apostrophe (same in line 307)
* Table 1: #3 missing closing bracket for while loop
* line 370: probably wrong cite command, should be \citep in my opinion
* line 352, second column: "(James, 1890). and" - dot should be removed
* line 371, second column: "than 0.3 or a scale" - probably "on" instead of "or"
* line 655: "Section 2a" - document reference points to Figure 2a instead
* line 622 to 626: there is content missing here - sentences are not complete
* line 764: "(1971 - 1855)" - birth year after year of death; according to Wikipedia, it is actually 1770–1850
* references:
  * inconsistent use of title case for the titles
  * "A general theoretical paradigm to understand learning from human preferences" - missing place of publication
  * "Constitutional ai: Harmlessness from ai feedback" - missing place of publication
  * "EVINCE: Optimizing Adversarial LLM Dialogues via Conditional Statistics and Information Theory" - cited differently than the other arXiv papers
  * "Deep reinforcement learning from human preferences" - missing place of publication; cited twice
  * "GPT-4 Technical Report" - cited differently than the other arXiv papers
  * "Training language models to follow instructions with human feedback" - missing place of publication
  * "Defining and characterizing reward hacking" - missing place of publication
  * "Learning to summarize from human feedback" - cited differently than the other arXiv papers

**Other Strengths And Weaknesses:**

* introduction:
  * "Users have reported that optimizing one task on ChatGPT degrades performance in others (Kirkpatrick et al., 2017; Rusu et al., 2015)." - ChatGPT was released in 2022, so how can a paper from 2017 be relevant? Even GPT-1 was just released in the same year as the Kirkpatrick article, which contains no mentions of GPT.
  * very inconsistent: sometimes the LLM is used for knowledge generation, later for processing
* 2.4 just reiterates content from the introduction with two additional references
* 3.2: I found the description confusing as the BEAM spectra and the letter spectrum partially contain the same terms such as joy.
* 3.3, revised statement: does not actually contain "newcomer" as latter is alleged
* paragraph at line 360-370 seems like it could breach anonymity
* no prompts
* no impact statement -> should be there, especially for such a topic

**Questions For Authors:**

* introduction: The motivational examples are not clear: Why are they relevant?
* 3.4: How does table 1 represent ERIS, if it shows the algorithm how the interaction between DIKE and ERIS is implemented?
  * How ERIS works is not shown at all?
  * How can ERIS be customized?
* evaluation: If there are 54+12=66 data samples and 24 are reserved for testing, how can Figure 2b illustrate the results for 54 letters?
* line 350-353, second column: "This concept is supported by Deisseroth’s optogenetic studies (Deisseroth, 2015), discussed in William James’ “The Principles of Psychology” (James, 1890). and corroborated in Minsky’s “Society of Mind” (Minsky, 1988)." - How can an article from 2015 can be discussed in 1890 and 1988 respectively? But maybe I am misreading the sentence.
* 4.2: How can a higher prediction entropy be an indicator that DIKE performs better than Zero-Shot GPT4, but an even higher entropy for humans is a cause of concern?
* 4.3, second paragraph: I thought that the guardrails, i.e. the constraints, are used to identify documents outside the targeted norm. DIKE is only involved in that process to a certain degree.
* What are the intended applications? Just aligning the LLM generation? Content moderation?
* How can DIKE be used as classification system and also as an agent in the adverserial process?
* How are ethical principles encoded into a spectrum of emotions?
* To what does E.1 related to? Maybe Figure 1, however there are seven rows and here are only six rows discussed.

**Relation To Broader Scientific Literature:**

I am not sure, whether this frameworks actually represents anything novel. It applies some insights from psychology to the alignment of AI generation, but the used terms are not very precise, so it makes it harder to understand the paper and the framework. Applications of the framework are only discussed in passing or not at all.

**Theoretical Claims:**

The article does not use much theory, which therefore can not be checked.

---

> ### Author Rebuttal · Authors · 2025-03-27
>
> Dear Reviewer,
> We respectfully request that you reconsider some of your comments, as they appear to misinterpret key aspects of our paper. For a constructive dialogue to proceed, we would like to address nine samples of your misunderstandings (there are more but space limited):
>
> 1. Relevance of a 2017 paper to ChatGPT
> Your comment: "ChatGPT was released in 2022, so how can a paper from 2017 be relevant?"
>
> This comment misunderstands how scientific citations function. The 2017 paper by Kirkpatrick et al. is cited for its work on catastrophic forgetting—a general neural network problem—not specifically in relation to ChatGPT. This represents a foundational concept in continual learning and is entirely valid as background theory.
>
> 2. Chronological misinterpretation
> Your comment: "How can an article from 2015 be discussed in 1890 and 1988 respectively?"
>
> This appears to be a misreading of a clear sentence. Our text states: "This concept is supported by Deisseroth (2015), discussed in James (1890), and corroborated by Minsky (1988)"—indicating that the idea appears in earlier and later works. We are not suggesting that James discussed Deisseroth's work. This misinterpretation raises concerns about the thoroughness of the review.
>
> 3. Context-dependent interpretation of entropy
> Your comment: "How can higher entropy be good for DIKE but bad for humans?"
>
> The paper explicitly explains this distinction. Higher entropy for DIKE demonstrates that it does not collapse into a single interpretation (unlike GPT-4's zero-shot approach), enabling better expressiveness. For human annotations, excessive variance reflects subjectivity and inconsistency. The interpretation of entropy differs depending on context and purpose.
>
> 4. ERIS functionality
> Your comment: "How does Table 1 represent ERIS, if it shows the algorithm how the interaction between DIKE and ERIS is implemented? How ERIS works is not shown at all?"
>
> Table 1 specifically describes the adversarial process between ERIS and DIKE in detail—including subtopic breakdown, iterative rebuttal, and modulation of contentiousness. Section 3.4 further elaborates on ERIS's cultural role.
>
> 5. Impact statement requirement
> Your comment: "No impact statement — should be there"
>
> Though we did not mark our impact statement in the end of Section 1, IT IS THERE, clearly, beneath our contribution summary.
>
> 6. Ethical concerns
> Your comment: "This could even advance towards censorship or forced reformulation of messages before publication."
>
> This represents a speculative ethical concern without grounding in the paper's actual scope. Our system is designed for LLM alignment—not mass surveillance or state censorship. The extrapolation into dystopian scenarios based on theoretical misuse extends beyond the scope of a scientific review. Current RLHF approaches could be similarly scrutinized according to your sentiment. You raised ethical concerns on this work, and if accurate, they should have applied equally to RLHF performed by all LLMs and hundreds of prior papers working on ethic alignment.
>
> 7. Ground truth and baselines
> Your comment: "No ground truth or baseline for Figure 2"
>
> Ground truth was clearly defined in Section 4.2: five human annotators plus GPT-4 and Gemini were used to produce averaged reference judgments. This methodology was explicitly described in the paper.
>
> 8. Applications
> Your comment: "What are the applications?"
>
> Applications are discussed explicitly throughout the paper: content moderation and ethical alignment. Sections 1, 3.2, and 5 specifically mention these applications, including escalation procedures to human reviewers. Ethical alignment represents a critical research area in contemporary AI development.
>
> 9. Map behaviors to emotions
> Your comment: "How are ethical principles encoded into a spectrum of emotions?"
>
> Well, Section3 provides the method thoroughly.  Please refer to the first reviewer's comments that this mapping process is solidly grounded in psychology, cognitive science, and self-supervised learning:
>
> By reviewer LkWg:
> "The paper establishes theoretical validity through psychological foundations (Plutchik/Scherer emotion models [1,2]) and cognitive-linguistic theories (James-Lange [3], Schachter-Singer [4])... Key theoretical claims about emotion-behavior mapping (BEAM's linear emotion spectra) and architectural separation benefits (preventing catastrophic forgetting through independent components) derive logical support from cited neuroscience (Deisseroth's optogenetics [5]) and machine learning principles (Kirkpatrick's catastrophic forgetting studies [6])."
>
> 10. Editorial comments
> Thank you for your editorial comments.  We will fix relevant ones promptly.
>
> We hope this clarification addresses your concerns and enables a more productive dialogue regarding the substantive aspects of our research.

---

### Official Review · Reviewer_LkWg · 2025-03-13

**Overall Recommendation:** 3

**Summary:**

This work introduces a novel three-branch framework (LLM/DIKE/ERIS) for ethical alignment in LLMs, inspired by governmental checks-and-balances. It decouples knowledge generation from ethical oversight and integrates emotion-driven behavioral modeling (via BEAM) with adversarial cultural adaptation. The framework aims to address RLHF’s limitations (e.g., catastrophic forgetting, cultural rigidity) by enabling interpretable, context-sensitive ethical reasoning. Pilot experiments on love letters demonstrate feasibility, with DIKE improving behavior classification accuracy by 11.3% over GPT-4 zero-shot.

**Claims And Evidence:**

1. The notion that separating knowledge generation (LLM) from ethical oversight (DIKE) prevents catastrophic forgetting finds support in references to [1] and a notable 11.3% improvement in behavior classification accuracy over GPT-4 through pilot tests on love letters. However, it faces criticism due to the absence of a direct RLHF comparison using QA benchmarks, experiments focused on sentiment analysis that avoid knowledge-intensive tasks, and a notable absence of an ablation study to isolate DIKE's impact on catastrophic forgetting.

2. The claim that self-supervised learning reduces bias compared to RLHF is weakly supported by DIKE's design, lacking in-depth bias analysis and fairness metrics. Additionally, there is no examination of GPT-4's inherent biases, such as gender stereotypes in rewritten texts.

3. The paper claims that BEAM enables precise emotion-behavior mapping for ethical adjustments, with evaluation results on love letter datasets providing support. However, the data distribution may be too simplistic to reflect real-world scenarios. The authors should consider conducting studies on more complex ethical scenarios, such as hate speech detection, to strengthen their claims.

[1] Kirkpatrick, J. et al. Overcoming catastrophic forgetting in neural networks. Proceedings of the National Academy of Sciences,114(13):3521–3526, 2017.

**Essential References Not Discussed:**

The paper covers most relevant literature.

**Experimental Designs Or Analyses:**

The experimental design demonstrates methodological coherence through three key validation strategies:
1. Emotion-behavior correlation analysis using GPT-4-generated rewrites to establish BEAM's mapping validity.
2. Comparative classification accuracy tests (DIKE vs GPT-4 vs humans) with inter-annotator agreement analysis to quantify behavioral understanding improvements.
3. Entropy measurements revealing DIKE's superior nuance in emotion spectrum modeling.

While constrained by LLM content policies necessitating love letter substitutions for hate speech evaluation, the design compensates through psychological plausibility checks analyzing historical correspondence to verify multi-emotion coexistence patterns. However, the evaluation shows three limitations:
1. Cultural adaptation claims rest on architectural potential rather than cross-cultural dataset validation.
2. Small test set size (24 letters) reduces statistical power for behavioral classification metrics.
3. Absence of direct RLHF performance comparisons leaves framework advantages partially theoretical.
4. The evaluation does not cover more complex or challenging scenarios.

**Methods And Evaluation Criteria:**

The proposed three-branch framework effectively addresses the limitations of RLHF by introducing a structured separation of responsibilities: ethical oversight (DIKE), adversarial cultural adaptation (ERIS), and knowledge generation (LLMs). This separation is complemented by BEAM's emotion-behavior mapping, which quantifies linguistic patterns through self-supervised learning. The evaluation criteria are well-suited to the problem, combining quantitative metrics—such as an 11.3% improvement in classification accuracy over GPT-4 and higher prediction entropy, indicating a more nuanced understanding of behaviors—with qualitative validation. However, the study needs broader cultural validation rather than focusing solely on the easier love letters scenarios.

**Other Comments Or Suggestions:**

No

**Other Strengths And Weaknesses:**

No

**Questions For Authors:**

No

**Relation To Broader Scientific Literature:**

The paper effectively underscores the limitations of RLHF, which, while enabling models to align with human preferences, often results in the models losing sight of their originally optimized responses. This persistent challenge is addressed in the paper through a novel approach that entirely decouples behavior modeling from knowledge representation, marking a significant departure from previous methods.

**Theoretical Claims:**

The paper establishes theoretical validity through psychological foundations (Plutchik/Scherer emotion models [1,2]) and cognitive-linguistic theories (James-Lange [3], Schachter-Singer [4]), rather than formal mathematical proofs. Key theoretical claims about emotion-behavior mapping (BEAM's linear emotion spectra) and architectural separation benefits (preventing catastrophic forgetting through independent components) derive logical support from cited neuroscience (Deisseroth's optogenetics [5]) and machine learning principles (Kirkpatrick's catastrophic forgetting studies [6]). While the adversarial DIKE-ERIS interaction draws conceptual strength from governmental checks-and-balances theory, the paper doesn't formally prove the framework's convergence properties or stability guarantees. The core theoretical contribution lies in the psychologically grounded architectural design rather than mathematical formalism.

[1] Plutchik, R. A psychoevolutionary theory of emotions. Social Science Information, 21(4-5):529–553, 1982.

[2] Scherer, K. R. What are emotions? and how can they be measured? Social Science Information, 44:693–727, 2005. doi: 10.1177/0539018405058216.

[3] James, W. What is an emotion? Mind, 9(34):188–205, 1884. URL http://www.jstor.org.proxy.lib.sfu.ca/stable/2246769.

[4] Lange, C. G. The emotions: A psychophysiological study. William & Wilkins, 1885.

[5] Deisseroth, K. Optogenetics: 10 years of microbial opsins in neuroscience. Nature Neuroscience, 18(9):1213–1225, 2015

[6] Kirkpatrick, J. et al. Overcoming catastrophic forgetting in neural networks. Proceedings of the National Academy of Sciences,114(13):3521–3526, 2017.

---

> ### Author Rebuttal · Authors · 2025-03-27
>
> We appreciate the reviewer's thoughtful feedback. Below, we address the main concerns raised.
>
> A. Complex Emotions and Emotion Modeling
> We acknowledge the critique regarding our treatment of complex emotions (e.g., pride, guilt, forgiveness). As discussed in Appendix D, we recognize the challenges in modeling these emotions, particularly their cultural variability and layered interpretations. Our approach deliberately begins with basic emotions to establish a clear and reliable foundation before extending to more complex emotional constructs.
>
> We distinguish that while basic emotions are largely unconscious and reflexive, complex emotions such as regret involve conscious deliberation. As noted in the paper, we propose modeling complex emotions as conscious behaviors. Given space constraints, we presented our preliminary work while acknowledging limitations and opportunities for future research.
>
> B. Data Scope and Self-Supervision Validity
>
> Regarding concerns about using GPT-4 to generate behavioral training data, we implemented several measures to mitigate potential feedback loop risks:
>
> 1. We incorporated rich stylistic diversity in our dataset (spanning over 50 authors across 200 years)
> 2. We conducted multi-agent validation using GPT-4, Gemini, and five human annotators (Section 4.2)
>
> We initially attempted to use hate speech datasets for our research, but encountered significant challenges: (1) none of the available datasets proved comprehensive, and (2) LLMs refused to process this data, citing policy violations. In today's increasingly polarized environment, using cultural or religious subjects for experiments risks inciting backlash. Our own experience attempting to publish a debate paper on a political issue resulted in negative emotions from university colleagues. We made earnest attempts to explore these avenues but ultimately had to reconsider our approach due to these practical constraints.
>
> We view self-supervised learning as advantageous for training data scalability. As demonstrated in Section 4.2, our approach can integrate human feedback, particularly within the ERIS module—a novel contribution of this work that provides context-based interpretation of ethical constitutions.
>
>
> C. Practicality and User Adaptability
>
> Our three-branch framework offers modularity and configurability:
>
> ERIS operates on-demand rather than continuously, reducing computational burden
> Guardrails are adjustable for domain or regional policies (Section 3.2)
> ERIS incorporates cultural variation, and DIKE's parameters can be customized by users or administrators
>
> While ethical oversight introduces some complexity, we believe the benefits of transparency, cultural flexibility, and safety justify this cost for sensitive applications.
>
> Regarding computational efficiency, our approach is comparable to current practices. LLMs already perform constitution checking (DIKE's function); we primarily add the ERIS judicial module for context-dependent interpretation, increasing token usage by at most 2x.
>
> Recent LLM developments (e.g., DeepSeek, GPT-4o, Claude 3.7) employ similar iterative validate-rethink loops, analogous to our DIKE-ERIS architecture. As NVIDIA CEO Jensen Huang noted at the March 18th GTC keynote, contemporary LLM architectures now routinely consume up to 20x more tokens and 150x more computation to improve reasoning quality and reliability. This suggests that enhancing AI quality necessarily incurs computational costs, a pattern observed since AlexNet.
>
> D. Theoretical Foundations
> We appreciate the recognition that our work establishes theoretical validity through psychological foundations (Plutchik/Scherer emotion models [1,2]) and cognitive-linguistic theories (James-Lange [3], Schachter-Singer [4]), rather than formal mathematical proofs.
>
> This interdisciplinary approach is indeed intentional. Many successful computational models, including CNNs, originated from neuroscience before their mathematical frameworks were fully developed. Before backpropagation and AlexNet demonstrated the validity of data-centric approaches, CNNs faced criticism for being computationally intensive and mathematically trivial.
> Our paper argues that while RLHF has strong mathematical foundations, it tends to address bias mitigation in a "whack-a-mole" fashion rather than drawing from behavioral science. Our theoretical foundation in psychology and cognitive science serves as inspiration for developing quantitative models.
>
> The complementary nature of interdisciplinary research strengthens our approach. Our quantitative model and unsupervised learning pipeline address training-data scalability challenges. We acknowledge there is much work to be done to validate and improve these methods, and we appreciate the opportunity to share these ideas with the research community.

---

> > ### Comment · Reviewer_LkWg · 2025-04-08
> >
> > Thank you for your rebuttal. Most of my concerns have been addressed. However, since the method proposed in the paper has only been tested on a small scale and lacks large-scale validation, I can only slightly increase my score.

---

### Decision · Program_Chairs · 2025-05-01

**Decision:**

Accept (poster)

**Comment:**

This work proposes a three-branch framework (LLM/DIKE/ERIS) for ethical alignment in LLMs, inspired by governmental checks-and-balances. Four reviewers reviewed this paper. Thanks to the authors' rebuttal, three reviewers are positive about this paper. Personally, the AC thinks the Ethical Alignment of Large Language Models is very important, and this paper provides a good try. Therefore, the AC recommends acceptance and requests the authors take the reviewers' detailed comments and the rebuttal into consideration in the final version.